# DETECTING DISTILLATION DATA FROM REASONING MODELS

## ABSTRACT

Reasoning distillation has emerged as an efficient and powerful paradigm for enhancing the reasoning capabilities of large language models. However, reasoning distillation may inadvertently cause benchmark contamination, where evaluation data included in distillation datasets can inflate performance metrics of distilled models. In this work, we formally define the task of **distillation data detection**, which is uniquely challenging due to the partial availability of distillation data. Then, we propose a novel and effective method *Token Probability Deviation* (*TBD*), which leverages the probability patterns of the generated *output* tokens. Our method is motivated by the analysis that distilled models tend to generate near-deterministic tokens for seen questions, while producing more low-probability tokens for unseen questions. Our key idea behind TBD is to quantify how far the generated tokens' probabilities deviate from a high reference probability. In effect, our method achieves competitive detection performance by producing lower scores for seen questions than for unseen questions. Extensive experiments demonstrate the effectiveness of our method, achieving an AUC of **0.918** and a TPR@1% FPR of **0.470** on the S1 dataset.

## 1 INTRODUCTION

Large Reasoning Models (LRMs) have shown impressive performance on complex tasks like mathematical reasoning and coding problems (Jaech et al., 2024; Guo et al., 2025; Yang et al., 2025; xAI, 2025). By articulating intermediate steps via Chain-of-Thought (CoT), LRMs dynamically allocate extra compute to challenging problems. However, such reasoning capabilities are typically limited to LRMs exceeding 100 billion parameters, hindering practical deployment in resource-constrained settings (Wei et al., 2022). To address this, recent studies have explored reasoning distillation, transferring reasoning abilities from LRMs to Small Language Models (SLMs) by simulating reasoning traces (Chen et al., 2025; Ye et al., 2025; Muennighoff et al., 2025b; Liu et al., 2025). This paradigm has been widely applied in cutting-edge models, such as DeepSeek R1 series (Guo et al., 2025), Sky-T1-32B-preview (Team, 2025), and Bespoke-32B (Labs, 2025).

In reasoning distillation, current methods generate reasoning trajectories and answers from LRMs for domain-specific questions, using these to supervise SLM training (Wu et al., 2025b; Li et al., 2025). Yet, the lack of transparency regarding distillation data raises concerns about benchmark contamination, where evaluation data inadvertently included in training can inflate performance metrics (Oren et al., 2024a; Xu et al., 2024a). These issues highlight the need to detect distillation data for distilled SLMs, ensuring transparency and fairness. Different from training data detection (Shi et al., 2024; Zhang et al., 2025b), the unique challenge of this task lies in partial availability: only the question is available at detection, without access to corresponding reasoning trajectories and answers. Accessing question-response pairs is generally infeasible due to the non-deterministic generation process in solution construction (Ye et al., 2025; Wu et al., 2025a) and the proprietary nature of datasets (Guo et al., 2025; Yang et al., 2025). Consequently, existing methods operating on input sequences struggle to obtain reliable membership signals given partial sample information.

In this study, we demonstrate that tokens generated by distilled models can expose information for identifying their distillation data. We observe that distilled models generally exhibit different probability distributions for members (*i.e.*, seen questions) and non-members (*i.e.*, unseen questions) in greedy decoding. In particular, distilled models tend to generate near-deterministic tokens for

Figure 1: **Overview of distillation data detection**. The top panel illustrates the pipeline of the reasoning distillation that distils the reasoning capacities of LRMs to smaller LLMs. The bottom panel illustrates the process of detecting distillation data.

members, while producing more low-probability tokens for non-members. This difference in token generation behavior indicates that the probability distributions of distilled models can be used to determine whether a given question was seen in the distillation process.

Inspired by the analysis, we propose a simple yet effective method – Token Probability Deviation (dubbed **TBD**[1]), which detects the distillation data through the probabilities of generated tokens, instead of input tokens. Our key idea behind TBD is to quantify how far the probabilities of generated tokens are from being fully deterministic. In particular, this can be accomplished by measuring the deviation of the generated token's probability from a high reference probability. In effect, our method produces smaller scores for members than for non-members at test time. By way of our method, we can achieve a clear separability of scores between seen and unseen questions, even when only the question component of each sample is available.

Empirically, we perform extensive experiments to validate the effectiveness of the proposed method across diverse models and various datasets, including S1, S1.1 (Muennighoff et al., 2025a) and LIMO (Ye et al., 2025). The results demonstrate that our method can significantly achieve superior performance than existing methods for detecting distillation data (See Table 1). For example, our method achieves an AUC of 0.918 and a TPR@1%FPR of 0.470 on the distilled model obtained by fine-tuning Qwen2.5-32B-Instruct on the S1 dataset, indicating the effectiveness of our method for detecting distillation data. Moreover, the ablation study shows that components in our method contribute to the overall high performance. In addition, experimental results show the robustness of our method across various datasets and models, enabling us to deploy our algorithm without task-specific hyperparameter tuning. In summary, our method shows superior performance in both the AUC and TPR@1%FPR metrics, showing the practicality of our method in real-world applications.

Our contributions and findings are summarized as follows:

- We first present the problem of distillation data detection and emphasize its unique challenge of partial availability. We then analyze the limitations of existing methods in the task of detecting distillation data.

- We propose Token Probability Deviation (dubbed **TBD**), a novel and effective method for detecting distillation data. The core idea of our method is to measure the deviation of generated tokens' probabilities from a high reference probability.

- We empirically show that our method can significantly outperform baselines for detecting distillation data, through extensive experiments conducted on various models and datasets.

---

[1]We denote the probability by B in our method, referred to as TBD.

## 2 PRELIMINARIES

**Reasoning distillation.** Reasoning distillation transfers the step-by-step reasoning behavior of large reasoning models (LRMs) into a smaller student language model by imitating the reasoning trajectories generated by teacher models (Guo et al., 2025; Li et al., 2025). Let $q$ denote a question drawn from a large-scale corpus $\mathcal{Q}$, collected from diverse sources. Using $q$ as a prompt, developers usually use LRMs to generate reasoning trajectories $c$ along with the final answer $a$ (See Appendix B.1 for an example). To construct a high-quality distillation dataset $\mathcal{D} = \{(q_i, c_i, a_i)\}_{i=1}^{N}$, developers then execute a meticulous selection process from an initial large-scale pool of candidates. The goal of reasoning distillation is to obtain a distilled model by fine-tuning an SLM on the resulting distillation dataset (See Figure 1). Formally, the objective of training can be formulated as:

$$\mathcal{L}_\theta = \sum_{t=1}^{N} \log P_\theta(y_t \mid y_{<t}, q), \tag{1}$$

where $q$ denotes the input question, and $y = \{y_1, y_2, \ldots, y_N\}$ represents the corresponding target sequence, comprising the reasoning trajectory $c$ and the final answer $a$. $P_\theta(y_t \mid y_{<t}, q)$ denotes the predicted probability of model for token $y_t$, given preceding tokens. This paradigm explicitly trains the student model to reproduce intermediate reasoning, aiming to internalize not just the outcomes but the procedural patterns of teacher models.

**Membership inference.** Membership Inference Attacks (MIAs) aim to predict whether a particular record is included in the training data (Shokri et al., 2017). MIAs are often used as a measure of information leakage, such as privacy disclosure (Mozes et al., 2023), copyright violations (Chang et al., 2023), and test set contamination (Xu et al., 2024a; Choi et al., 2025). The definition of *traditional MIAs* is as follows: Given a trained model $f(\boldsymbol{x}, \boldsymbol{\theta})$ and a data point $(\boldsymbol{x}, y)$, an attacker infers whether a target data point belongs to the training data $\mathcal{D}_{train}$. In traditional MIA settings, they often require strong assumptions, such as training multiple shadow models and accessing to the underlying data distribution. This is often impractical for LLMs due to the unavailability of training data distribution and high training costs. The existing MIAs on LLMs usually aim to determine whether a given piece of text $\boldsymbol{x}$ is part of the training dataset for a large language model $\mathcal{M}$, by computing a membership score $\mathcal{S}(\boldsymbol{x}, \mathcal{M})$. Training data detection methods for LLMs generally design a scoring function that computes a score for each input (Li, 2023; Shi et al., 2024; Zhang et al., 2025b). Although some methods for pretraining and fine-tuning data detection have been studied, membership inference on distillation data for reasoning distillation remains underexplored. In the next section, we introduce the distillation data detection task, a tailored formulation of this problem.

## 3 DISTILLATION DATA DETECTION

In this section, we formally define the ***Distillation Data Detection*** task, which is uniquely challenging due to the partial availability of distillation data. The goal of our task is to predict whether a given question is included in the model's distillation dataset.

**Problem definition.** Using question $q$ as a prompt, developers often generate training data by sampling responses from multiple advanced LRMs and by refining them to obtain high-quality reasoning trajectories $c$ and corresponding answer $a$ (Ye et al., 2025; Wu et al., 2025b; Tian et al., 2025; Zhuang et al., 2025). However, the resulting distillation dataset $\mathcal{D} = \{(q_i, c_i, a_i)\}_{i=1}^{N}$ is often proprietary (Guo et al., 2025; Yang et al., 2025)—*i.e.*, the exact reasoning trajectory and answer are inaccessible for a given question. Also, due to the non-deterministic generation process and post-hoc filtering, it is generally infeasible to recover the exact reasoning trajectory or answer associated with a given question. Thus, we study a more practical *question-only* setting in which an auditor can query a distilled model $\mathcal{M}$ with question $q$ and obtain model outputs, but has no access to the corresponding reasoning trajectories $c$ and answer $a$ of a datapoint.

Formally, let $\mathcal{Q}_\mathcal{D} = \{q_i : (q_i, c_i, a_i) \in \mathcal{D}\}$ denote the set of questions from a distillation dataset used for training base models. We pose distillation data detection as a level-set estimation problem defined on a scoring function $\mathcal{S}(q, \mathcal{M})$ as:

$$G(q; \mathcal{M}) = \begin{cases} 1 & \text{if } \mathcal{S}(q, \mathcal{M}) < \lambda, \\ 0 & \text{if } \mathcal{S}(q, \mathcal{M}) \geq \lambda, \end{cases} \tag{2}$$

where $G = 1$ indicates *member* ($q \in \mathcal{Q}_\mathcal{D}$) and $G = 0$ indicates *non-member* ($q \notin \mathcal{Q}_\mathcal{D}$), with $\lambda$ being a case-dependent threshold. The key difficulty of this task lies in ***partial availability***: the training datapoint is triple $x = (q, c, a)$, yet only the question $q$ is available at test time. Consequently, the design of $\mathcal{S}$ must rely solely on question-conditioned behaviours of $\mathcal{M}$, rather than on likelihood-driven metrics over the ground truth $(c, a)$.

**Challenge of partial availability.**   Most prior work on training data detection for LLMs assumes access to the *entire* training sample, which contains complete information seen during training (Mattern et al., 2023; Fu et al., 2024; Mireshghallah et al., 2022). In this setting, the scoring function $\mathcal{S}$ can be defined directly in terms of sample likelihoods, exploiting probability estimates over input tokens. Existing training data detection approaches targeting LLMs typically leverage a scoring function that computes a score for each input sequence. For example, MIN-K% (Shi et al., 2024) computes the average log-likelihood of the lowest K% tokens scores over the input, effectively using low-probability tokens as signals of membership. Such approaches are well-suited when full sample information is observable at detection time.

However, these approaches may perform poorly in the setting of distillation data detection, where distillation data are only partially available. The absence of joint question-response pairs weakens the key signal exploited by likelihood-based approaches operating over input sequences, leaving them ill-suited for this task. To illustrate, we analyze the distribution of MIN-K% scores for member versus non-member questions. As shown in Figure 2, the two distributions exhibit substantial overlap, indicating limited separability and poor discriminative power when only questions are available. This highlights the requirement for alternative scoring functions effective under partial availability. Motivated by the challenge of partial availability, we investigate whether the *token generation behavior* of distilled models, conditioned solely on $q$, can serve as a reliable signal of membership.

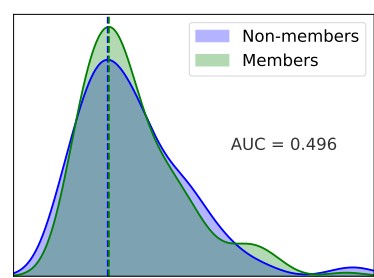

Figure 2: Scores distribution of Min-K% for members and non-members, obtained from the distilled model trained on the LIMO dataset using the Qwen2.5-32B-Instruct base model.

## 4   METHOD

To address the challenge of partial availability, we explore a question-only scoring approach that leverages the token-level generation behavior of distilled reasoning models. We begin by comparing the probability patterns of tokens between member and non-member questions (See Section 4.1). Then, building on our empirical observations, we propose *Token Probability Deviation* (TBD), a simple yet effective method to detect distillation data (See Section 4.2).

### 4.1   MOTIVATING ANALYSIS

**Analysis setup.**   The goal of our analysis is to investigate whether the token-level probability patterns produced by a distilled model differ between member and non-member questions. Following prior work (Muennighoff et al., 2025b), we first distill the reasoning capabilities to the Qwen2.5-32B-Instruct model via supervised full-parameter fine-tuning on the S1 dataset. The dataset is split into training and testing subsets with an 8:2 ratio, from which we sample members (training set) and non-members (testing set). This ensures an i.i.d setup, with both groups drawn from the same underlying data distribution. For each question, we generate a response from the distilled model using greedy decoding and extract the token probabilities of the response for comparison. Example question prompts are provided in Appendix B.2.

**Members generate near-deterministic tokens more frequently.**   To examine distributional differences in token probabilities, we analyze sequences of up to 300 generated tokens for both member and non-member questions. Figure 3a shows the token-wise maximum probability distributions from the distilled model across 20 member and 20 non-member samples. The horizontal axis denotes the token index, while the vertical axis reports the probability assigned to the corresponding

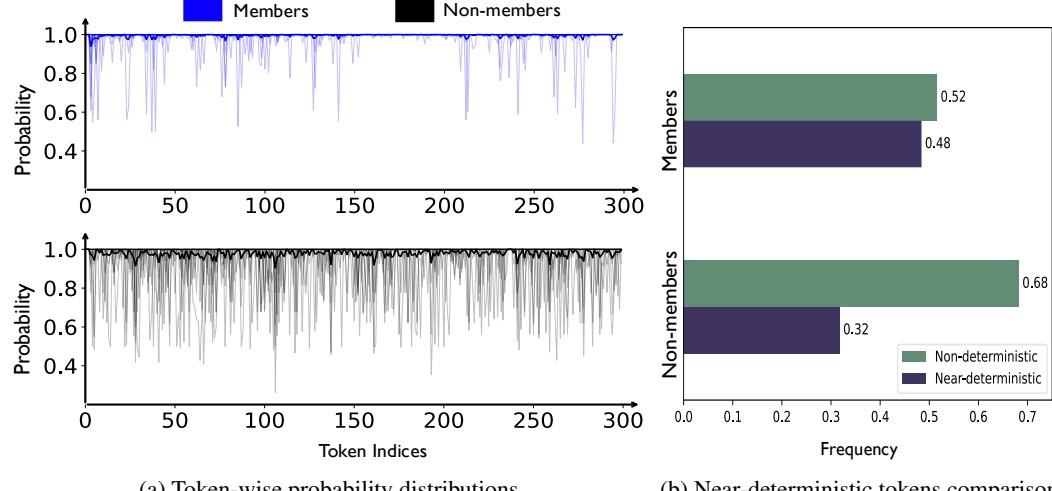

(a) Token-wise probability distributions

(b) Near-deterministic tokens comparison

Figure 3: Comparison of token-level generation behaviour of distilled models for 20 member and 20 non-member questions under greedy decoding. **(a) Token-wise probability distributions**: we contrast the distribution of token-wise probability between members and non-members, showing that non-members tend to produce more tokens with lower probability. **(b) Near-deterministic vs. non-deterministic tokens**: near-deterministic tokens denote generated tokens with probabilities approaching 1, and vice versa for non-deterministic tokens. The distilled reasoning model tends to generate more near-deterministic tokens for members.

token. Thin lines correspond to individual samples, and bold lines denote mean token-wise probability across all members or non-members. We observe that the distilled model tend to frequently generate tokens with probabilities close to 1 for members, while producing more low-probability tokens for non-members. Figure 3b further contrasts the frequency of near-deterministic tokens (with probability approaching 1) against non-deterministic tokens. The results show that the distilled model produces a substantially higher fraction of near-deterministic tokens for member questions. This suggests that generation probabilities are likely to carry membership signals, motivating our design of a scoring function that leverages tokens generated by the distilled model to detect distillation data. Building on these insights, we introduce *Token Probability Deviation* in the next section.

## 4.2 TOKEN PROBABILITY DEVIATION

Motivated by our preliminary analysis, we propose Token Probability Deviation (TBD), a method that exploits the observation that member questions tend to elicit near-deterministic tokens, whereas non-members induce relatively more low-probability tokens. Unlike many methods for detecting training data from LLMs, which rely on *input* token probabilities (Shi et al., 2024; Zhang et al., 2025b), TBD utilizes the *generated* token probabilities along the model's reasoning trajectory for given question $q$. This design sidesteps the partial-availability constraint of distillation datasets while providing a simple, model-agnostic signal for distillation data detection.

**Token Probability Deviation.** The core idea of our method is to measure the deviation of the generated tokens' probability from a high reference probability. Given a question entailed with a sequence of tokens $q = \{q_1, q_2, ..., q_N\}$, the tokens generated by model can be denoted as $y = \{y_1, y_2, ..., y_i\}$. We use $p_\theta(y_i \mid y_{<i}, q)$ to denote the probability that the target model predicts $y_i$, given the question prompt $q$ and the generated text prefix $y_{<i} = \{y_1, y_2, ..., y_{i-1}\}$. We first quantify the deviation between the probability of the generated token $y_i$ and the reference probability $\tau$. Formally, we define the deviation term as:

$$d_i(q; \tau) = \max\big(0, \tau - p_\theta(y_i \mid y_{<i}, q)\big), \tag{3}$$

where $\max(0, \cdot)$ ensures that only outlier tokens, whose probabilities are below the threshold $\tau$, contribute to the final score computation. Since distilled reasoning models tend to generate tokens with extremely high probability for seen questions, outlier tokens are likely to display a highly

distinctive membership signal. Therefore, by computing the deviation of the outlier tokens, we expect to obtain a more distinctive signal for data membership.

Considering that the earlier generated tokens are likely to be more representative of the behaviour of the model for members and non-members, we perform a truncation operation to focus on the first $M$ generated tokens. To obtain the final robust sentence-level score, we compute the average of the token's deviation $d_i(q; \tau)$. Concretely, the final score can be formulated as:

$$\mathcal{S}(q, \theta) = \frac{1}{E} \sum_{i=1}^{M} d_i(q; \tau)^{\alpha}, \tag{4}$$

where $E = \sum_{i=1}^{M} \mathbf{1}(p_\theta(y_i \mid y_{<i}, q) < \tau)$ denotes the number of outliers among the first $M$ tokens.

In practice, we introduce a tunable parameter $\alpha$ to adjust the contribution of tokens to the scoring function. For instance, a small value of $\alpha$ (e.g., 0.6) can amplify the deviation of a generated token's probability from $\tau$. Our experimental results in Figure 4b show that a suitable $\alpha$ can yield an improved TPR@1% FPR for detecting distillation data.

**Detection with token probability deviation.** Our method enables us to build a detector $G(q; M)$ for a distilled reasoning model to infer the membership of question $q$. In particular, our method is robust across various datasets, enabling us to deploy our algorithm without task-specific hyperparameter tuning. At test time, samples with lower scores $\mathcal{S}(q, \theta)$ are classified as distillation data and vice versa. By way of our method, we can obtain a clear distinction between seen questions and unseen questions, establishing excellent performance for detecting distillation data.

## 5 EXPERIMENTS

In this section, we evaluate the performance of our method across several datasets with multiple models of different sizes. Extensive experiments demonstrate the effectiveness of our method, which designs a scoring function using generated tokens instead of input tokens.

### 5.1 EXPERIMENTAL SETUP

**Datasets and models.** We conduct experiments on several high-quality distillation datasets provided by previous work, including S1, S1.1 (Muennighoff et al., 2025b) and LIMO (Ye et al., 2025). The details of datasets are provided in Appendix B.1. Specifically, we fine-tune the base model (e.g, Qwen2.5-32B-Instruct) on these datasets with a full-parameter supervised fine-tuning strategy. In addition, we also perform experiments on different-sized base models, such as Qwen2.5-7B-Instruct, Qwen2.5-14B-Instruct and Qwen2.5-32B-Instruct models (Qwen et al., 2025).

**Baseline methods.** We compare our method with current competitive baselines: (1) **Perplexity** (Li, 2023): uses the perplexity of input text as a metric. (2) **Zlib** (Carlini et al., 2021): computes the ratio of example perplexity and zlib compression entropy (3) **Lowercase** (Carlini et al., 2021): computes the ratio of the perplexity on the text before and after lowercasing. (4) **Neighbor** (Mattern et al., 2023): perturbs the input sentence with masked language models to create "neighbor" and compares the loss of the input sentence with the average loss of the neighbor sentences. (5) **Min-K%** (Shi et al., 2024): computes the average log-likelihood of K% outlier tokens with the smallest predicted probability. (6) **Min-K%++** (Zhang et al., 2025b): compares the probability of the target token with the expected probability of all tokens within the vocabulary. (7) **Infilling Score** (Raoof et al., 2025): computes the ratio of the infilling probability of the ground-truth token and the maximum causal likelihood token. These methods typically detect training data by designing likelihood-based scores derived from input tokens, while our method leverages tokens generated from models to detect training data. Additionally, we introduce two vanilla variants of our method, which use generated tokens to determine data membership. Specifically, (8) **Generated Perplexity**: computes the perplexity using the probabilities of generated tokens. (9) **Generated Min-K%**: computes the average log-likelihood of K% generated tokens with the lowest predicted probability.

**Implementation details.** To effectively evaluate our method, we fine-tune the Qwen2.5-32B-Instruct model separately on S1, S1.1, and LIMO distillation datasets to obtain diverse distilled

Table 1: AUC of our method and baselines on diverse distilled models. These models are produced through fine-tuning diverse different-sized models (e.g., Qwen2.5-32B-Instruct) on various distillation datasets, including S1, LIMO and S1.1 datasets. † indicates methods that compute score using output tokens. **Bold** shows the superior result.

| Method | Qwen2.5-7B-Instruct | | | Qwen2.5-14B-Instruct | | | Qwen2.5-32B-Instruct | | |
|---|---|---|---|---|---|---|---|---|---|
| | S1 | LIMO | S1.1 | S1 | LIMO | S1.1 | S1 | LIMO | S1.1 |
| *Input-token-based methods* | | | | | | | | | |
| Perplexity (Li, 2023) | 0.444 | 0.482 | 0.503 | 0.449 | 0.498 | 0.517 | 0.433 | 0.499 | 0.487 |
| Lowercase (Carlini et al., 2021) | 0.435 | 0.472 | 0.493 | 0.467 | 0.507 | 0.489 | 0.459 | 0.475 | 0.463 |
| Zlib (Carlini et al., 2021) | 0.474 | 0.486 | 0.467 | 0.467 | 0.495 | 0.471 | 0.448 | 0.496 | 0.447 |
| Neighbor (Mattern et al., 2023) | 0.539 | 0.503 | 0.441 | 0.543 | 0.500 | 0.435 | 0.555 | 0.503 | 0.444 |
| MIN-K% (Shi et al., 2024) | 0.443 | 0.480 | 0.494 | 0.453 | 0.496 | 0.509 | 0.437 | 0.496 | 0.479 |
| MIN-K%++ (Zhang et al., 2025b) | 0.472 | 0.458 | 0.486 | 0.509 | 0.508 | 0.489 | 0.461 | 0.461 | 0.439 |
| Infilling Score (Raoof et al., 2025) | 0.529 | 0.529 | 0.520 | 0.534 | 0.544 | 0.493 | 0.574 | 0.489 | 0.475 |
| *Output-token-based methods* | | | | | | | | | |
| Generated Perplexity† | 0.753 | 0.605 | 0.564 | 0.785 | 0.596 | 0.558 | 0.847 | 0.662 | 0.619 |
| Generated MIN-K† | 0.754 | 0.604 | 0.563 | 0.785 | 0.596 | 0.559 | 0.847 | 0.661 | 0.619 |
| Ours† | **0.855** | **0.694** | **0.617** | **0.870** | **0.671** | **0.562** | **0.918** | **0.728** | **0.649** |

reasoning models. For the main results, the original datasets are split into training and testing subsets, with an 8:2 train-test split. We then perform full-parameter fine-tuning on 8 A100 GPUs using DeepSpeed ZeRO-3 optimization, with a sequence length limit of 16,384 tokens. The details of training parameters are provided in the Appendix B.2. To ensure fair evaluation, we construct balanced datasets of member and non-member samples, drawn respectively from the training and test sets, ensuring an IID setting. For two vanilla variants of our method, *Generated Perplexity* and *Generated Min-K%*, the sample score is computed using only the first 1,000 generated tokens. For main experiments, we apply a greedy decoding strategy for generation, and compute the TBD score using the first 300 generated tokens with $\tau = 1$ and $\alpha = 0.6$.

**Evaluation metrics.** We evaluate the performance of our method and baselines for detecting distillation data by measuring the following metrics: (1) AUC, the area under the receiver operating characteristic curve; (2) TPR@1%FPR, the true positive rate at 1% false positive rate (Carlini et al., 2022). Instead of paying equal attention to members and nonmembers, this metric pays more attention to members and evaluates whether one can confidently identify members.

## 5.2 EXPERIMENTAL RESULTS

**Is our method effective across models trained on various datasets?** To investigate the performance of our method across diverse distilled reasoning models, we fine-tune the diverse models on three distillation datasets, including S1, LIMO and S1.1 datasets. Table 1 shows that our method significantly outperforms the baselines, achieving superior performance for detecting distillation data. We also present the TPR@1% FPR score of our method and baselines in Appendix C.2. Firstly, our experiments demonstrate that tokens generated by distilled models can serve as effective information for detecting distillation data. Furthermore, empirical evidence suggests that our method can detect distillation data even under a low false-positive rate constraint, showing the practicality of our method in real-world applications. For example, our method achieves a high AUC of 0.918 and a TPR@1% FPR of 0.470 on the distilled model obtained by fine-tuning Qwen2.5-32B-Instruct on S1. Overall, our experimental results demonstrate the effectiveness of our method for detecting distillation data across diverse models and datasets.

**Is our method effective across various models?** To investigate the effectiveness of our method across various models, we conduct experiments on the S1 dataset with three different LLMs, including Llama-3.1-8B-Instruct (Dubey et al., 2024), Gemma-7B-it (Team et al., 2024) and Mistral-7B-Instruct-v0.3 (Jiang et al., 2023) models. In Appendix C.2, we provide the AUC and TPR@1%FPR scores for our method and the baselines across various models. The results indicate that our method

 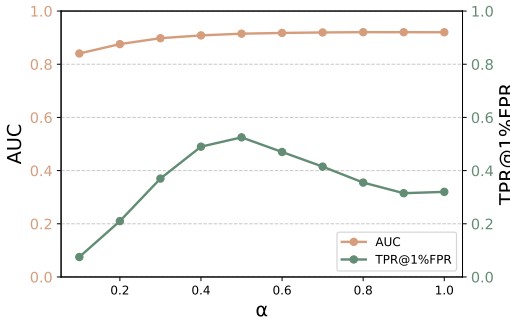

(a) AUC score vs. size of distillation data       (b) AUC and TPR@1%FPR scores vs. parameter $\alpha$

Figure 4: Effect of distillation data size (4a) and parameter $\alpha$ (4b) on our method's performance.

consistently achieves superior performance compared to baselines, demonstrating its model-agnostic nature and broad applicability.

**Is our method effective with models of different parameter sizes?**   To validate the effectiveness of our methods on distilled models of different sizes, we fine-tune Qwen2.5-7B-Instruct, Qwen2.5-14B-Instruct, and Qwen2.5-32B-Instruct models on various datasets, respectively. As presented in Table 1, the results show that our method achieves superior performance for detecting distillation data, demonstrating the effectiveness of our method across models of different sizes. Among three models of different sizes, our method achieves the best AUC and TPR@1%FPR on the S1 dataset under the 32B model. The properties of models and characteristics of datasets may potentially influence our method's performance.

**How does the distillation data size affect our method?**   To investigate the performance of our method on varying dataset scales, we conduct experiments on the S1 dataset and fine-tune Qwen2.5-32B-Instruct with data sizes of 600, 700, and 800. At test time, we construct a balanced dataset to evaluate the performance of our method. Figure 4a shows the AUC of our method with various sizes of distillation datasets. The results demonstrate our method consistently achieves reliable detection performance across diverse distilled models. In addition, we observe that the AUC of our method slightly increases with the size of the distillation dataset, likely because the distilled model trained on more data exhibits enhanced generation behaviour that improves detection. In summary, our experiment indicates the effectiveness of our method with distillation datasets of different sizes.

**How does $\alpha$ affect the performance of our method?**   Our method introduces a tunable parameter $\alpha$ to adjust the contribution of tokens on the sample score. For instance, a small value of $\alpha$ amplifies the deviation of a generated token's probability from $\tau$, thereby increasing its impact on the sample score when deviations are minor. We conduct experiments with varying $\alpha$ values to examine their effect on our method's performance, based on the distilled model fine-tuned from Qwen2.5-32B-Instruct on S1. Figure 4b shows the AUC and TPR@1%FPR scores of our method with varying $\alpha$. Note that setting $\alpha$ to 1 is equivalent to applying our method without deviation adjustment. Increasing $\alpha$ initially improves the AUC score, and performance ultimately stabilizes as $\alpha$ continues to increase. The TPR@1%FPR score significantly rises as $\alpha$ increases, reaching a peak near $\alpha = 0.6$, and subsequently decreases. This behaviour allows us to deploy our algorithm flexibly by simply adjusting $\alpha$, targeting the preferred metric in practical applications. Overall, our method can significantly improve TPR@1%FPR by applying a $\alpha$ in our method.

**How does truncation length $M$ affect the performance of our method?**   Our method introduces a truncation operation to compute the sample score using the first $M$ generated tokens. To study how the length of tokens affects the performance of our method, we perform an evaluation by adjusting the number of tokens used to compute the score. Concretely, we set the number of tokens from 50 to 1000 with a step size of 50. In our experiment, setting the truncation length to 300 corresponds to computing the score using the first 300 generated tokens. We evaluate our method on distilled models obtained by fine-tuning Qwen2.5-32B-Instruct on three distillation datasets, reporting the AUC and TPR@1%FPR of our method with different truncation lengths. The figure 5a shows that

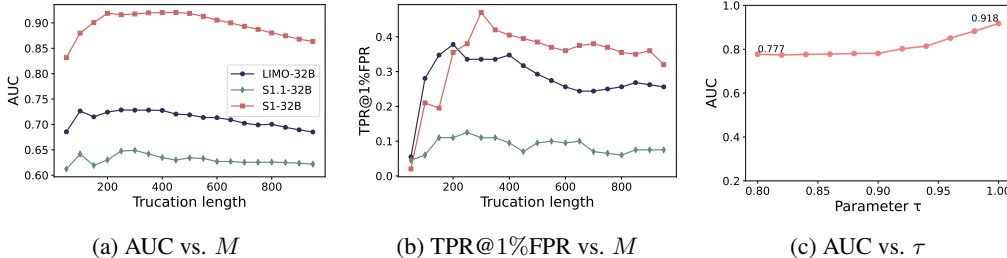

(a) AUC vs. $M$         (b) TPR@1%FPR vs. $M$         (c) AUC vs. $\tau$

Figure 5: Ablation study on hyperparameter $M$ and threshold $\tau$. We present AUC and TPR@1%FPR of our method with varying truncation length $M$ on three datasets (5a & 5b), and AUC of our method under varying threshold $\tau$ on the S1 dataset (5c).

Table 2: Ablation study on the components of our method with the distilled model fine-tuned from Qwen2.5-32B-Instruct on the S1 dataset. Note that excluding parameter $\alpha$ corresponds to setting $\alpha = 1$. The grey rows correspond to the final design of our method with different $\alpha$. **Bold** shows the superior result, with the runner-up underlined.

| Truncation length | Token deviation | Parameter $\alpha$ | AUC | TPR@1% FPR |
|:---:|:---:|:---:|:---:|:---:|
| ✗ | ✗ | ✗ | 0.847 | 0.170 |
| $M = 300$ | ✗ | ✗ | 0.903 | 0.290 |
| $M = 300$ | ✓ | ✗ | 0.920 | 0.320 |
| $M = 300$ | ✓ | $\alpha = 0.6$ | 0.918 | **0.470** |
| $M = 300$ | ✓ | $\alpha = 0.7$ | **0.920** | 0.415 |

the AUC initially increases with truncation length, but declines as the length continues to increase. Our method consistently reaches its optimal performance across various datasets when the truncation length is around 300. Similarly, Figure 5b show that the TPR@1%FPR score exhibits a similar trend. These findings indicate the robustness of our method across various datasets, allowing users to deploy our algorithm with a fixed truncation length value.

**Effect of threshold $\tau$ on the performance of our method.** Our method introduces a reference probability $\tau$ to quantify the deviation of the generated token's probability from the reference probability. In Figure 5c, we show the AUC of our method with varying $\tau$ on the distilled model obtained by fine-tuning Qwen2.5-32B-Instruct on S1. The result shows that our method achieves superior performance with a large value of $\tau$. As described in Section 4.1, the distilled reasoning model tends to generate tokens with extremely high probability. Thus, applying a high reference probability can help identify outlier tokens and achieve better performance for detecting distillation data.

**Decomposing the contribution of our method.** As described in Equation 4, our method can be decomposed into three key components: **(1)** truncation operation $M$, which truncates the generated sequence to the first $M$ tokens; **(2)** token deviation measure $d_i(q; \tau)$, which measures the deviation of the generated token's probability from a reference probability $\tau$; and **(3)** tunable parameter $\alpha$, which adjusts the contribution of tokens on the sample score. To elucidate individual contributions of each component, we conduct an ablation study in Table 2. We start by computing the average predicted probabilities over the first 1000 generated tokens. We then gradually incorporate the truncation operation, token deviation measure, and parameter $\alpha$ into the score computation, leading to the final formulation of our method. In particular, applying a truncation operation, using only the first 300 generated tokens for score computation, leads to a significant improvement in performance. Secondly, our method achieves better performance after applying the token deviation measure, indicating that focusing on outlier tokens produces a more distinguishable membership signal. Finally, we introduce a $\alpha$ to adjust the contribution of tokens to the score, leading to a significant improvement in the TPR@1%FPR score. By combining these components, we obtain the final formulation of our method, which achieves superior performance in both AUC and TPR@1% FPR scores.

Table 3: AUC scores of our method and baselines on the paraphrased S1 dataset, evaluated across different-sized models. **Bold** shows the superior result.

| Method | Qwen2.5-7B-Instruct | Qwen2.5-14B-Instruct | Qwen2.5-32B-Instruct |
|---|---|---|---|
| Perplexity (Li, 2023) | 0.463 | 0.468 | 0.469 |
| Lowercase (Carlini et al., 2021) | 0.503 | 0.493 | 0.543 |
| Zlib (Carlini et al., 2021) | 0.497 | 0.497 | 0.496 |
| Min-k% (Shi et al., 2024) | 0.469 | 0.475 | 0.473 |
| Min-k%++ (Zhang et al., 2025b) | 0.500 | 0.494 | 0.527 |
| Ours | **0.615** | **0.692** | **0.691** |

## 6 DISCUSSION

**How does partial availability affect the performance of our method and baselines?** To illustrate the scenarios where our method provides utility, we evaluate our method and baselines on the S1 dataset using the Qwen2.5-7B-Instruct model across three distinct settings:(1) using only the question, (2) using the question along with the reasoning trajectories, and (3) using the full sample comprising the question, reasoning trajectories and answer.

We provide the AUC scores of baselines and our method in Appendix C.2. The results show that baseline methods are effective in settings where reasoning trajectories and answers are available, while their performance notably degrades when only the question is available. The finding indicates that partial availability leads to poor performance of baselines in distillation data detection. The results show that our TBD is the only effective method in the Question-Only setting, while all baseline methods fail to detect distillation data. Our method enables effective detection in the realistic and challenging setting, where only the is available, achieving meaningful performance without relying on trajectories or answers.

**Is our method effective for question paraphrasing?** To examine the performance of our method under reasoning distillation with paraphrased questions, we conduct experiments on the S1 dataset across different models, including Qwen2.5-7B-Instruct, Qwen2.5-14B-Instruct and Qwen2.5-32B-Instruct models. We use GPT-5-mini[2] to paraphrase the original question, obtaining a rephrased version that remains semantically consistent with the original question. We then evaluate our method on paraphrased questions to simulate a scenario where the original questions used for reasoning distillation are unavailable. The Table 3 reports the AUC scores of baselines and our method. The results show that our method consistently outperforms baselines, indicating its capability to detect distillation data in the question paraphrasing scenario.

## 7 CONCLUSION

In this work, we first present the problem of distillation data detection and emphasize its unique challenge of partial availability. We propose Token Probability Deviation (*TBD*), a novel and effective method for detecting distillation data. Our method utilizes generated tokens instead of input sequences to identify data membership. This can be achieved by measuring the deviation of generated tokens' probabilities from a high reference probability. Experimental results show that our method is robust to parameter choice, enabling us to deploy our algorithm without task-specific hyperparameter tuning. In addition, our method can detect distillation data even under a low false-positive rate constraint, showing the practicality of our method in real-world applications. In summary, extensive experiments demonstrate the effectiveness of our method on various datasets across diverse models in distillation data detection. We hope that our study can advance further research on data contamination resulting from reasoning distillation.

**Limitations** Our work focuses on detecting training data used in reasoning distillation. Our method is limited in the scope of reasoning distillation with supervised fine-tuning, leaving other scenarios to be explored in future work.

---

[2]https://platform.openai.com/docs/models/gpt-5-mini

## ETHICS STATEMENT

Our work aims to detect distillation data, a data leakage problem resulting from reasoning distillation. The proposed methodology aims to identify data potentially used in reasoning distillation. Regarding data access, the distillation datasets we employed in our work come from prior research and do not involve privacy information. This paper presents work whose goal is to advance research on data contamination resulting from reasoning distillation. There are many potential societal consequences of our work, none of which we feel must be specifically highlighted here.

## REPRODUCIBILITY STATEMENT

We have made efforts to ensure that the experimental results in this paper are reproducible. We provide an anonymous link to the downloadable source code in the supplementary materials for others to reproduce the results in our experiments. The experimental setup, including training steps, datasets, models and hardware details, is described in detail in this paper. To support reproducibility, we provide detailed instructions on code execution for our experiments. We hope that our efforts can help other researchers reproduce our work and further advance the field.

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

# Appendix

## Table of Contents

## A  RELATED WORKS

In this paper, we propose a problem about distillation data detection, which is related to an amount of literature on reasoning distillation and detecting training data from Large Language Models (LLMs). We discuss related works in two directions relevant to our study.

**Distilling reasoning capability from LRMs.**  Large Reasoning Models (LRMs) exhibit remarkable performance in solving complex tasks, achieving this by training the model to produce a long chain of thought reasoning process before responding with the final answer (Jaech et al., 2024; Yang et al., 2025; Comanici et al., 2025). However, developing reasoning models, which achieve reasoning capacity compared to large reasoning models, remains a significant challenge for the research community (Kumar et al., 2024; Xin et al., 2024; Wan et al., 2024). Recently, a growing literature has focused on reasoning distillation, which improves the reasoning capabilities of models with lower computational cost (Guo et al., 2025; Yang et al., 2025; Wu et al., 2025b; Ma et al., 2025). In practice, distillation methods often employ supervised fine-tuning to enable a model to mimic reasoning trajectories generated by large reasoning models (Guo et al., 2025; Muennighoff et al., 2025b; Ye et al., 2025; Liu et al., 2025). For instance, S1 (Muennighoff et al., 2025b) and LIMO (Ye et al., 2025) enhance the reasoning capacity of small language models by fine-tuning models on well-crafted distillation datasets. However, training models on distillation datasets that overlap with benchmark data can inflate the performance of distilled models. Thus, our work aims to develop detection methods for identifying distillation data potentially used in reasoning distillation.

**Detecting training data from LLMs.**  Training data detection on LLMs has been studied in previous works, encompassing fine-tuning data detection and pretraining data detection (Mattern et al., 2023; Fu et al., 2024; Zhang et al., 2025b; Shi et al., 2024; Raoof et al., 2025). Training data may pose risks such as privacy leakage, where training data containing personal information may lead to privacy leakage (Grynbaum & Mac, 2023; Mozes et al., 2023). Furthermore, the training dataset may inadvertently include data from benchmarks, which compromises the reliability of benchmark evaluations (Oren et al., 2024b; Choi et al., 2025; Xu et al., 2024b). Fine-tuning data detection for LLMs aim to determine the training data used for fine-tuning (Mattern et al., 2023; Mireshghallah et al., 2022). The repeated exposure of the fine-tuning data across multiple training epochs increases their vulnerability to privacy attacks. Pretraining data detection aims to determine whether a piece of text is included in the pretraining dataset (Zhang et al., 2025a). The task is particularly challenging due to the massive scale of the pretraining corpus and the fact that pretraining usually runs for only one epoch (Shi et al., 2024). Previous studies often design scoring functions that compute a score for each input sequence to detect training data from LLMs (Zhang et al., 2025b). Our work proposes

a problem of distillation data detection and a novel and effective method, which leverages generated tokens instead of the input sequence to detect distillation data.

## B  EXPERIMENTAL DETAILS

### B.1  DETAILS OF DATASETS AND MODELS

**Datasets and Models.**  To obtain diverse distilled reasoning models, we fine-tune different models on three well-crafted distillation datasets, including S1, S1.1 (Muennighoff et al., 2025b) and LIMO (Ye et al., 2025). Use questions as prompts, developers often generate reasoning trajectories along with the final answer from advanced large reasoning models to construct distillation datasets. The S1 and S1.1 datasets contain 1,000 examples, whose reasoning trajectories are produced from Gemini (Comanici et al., 2025) and DeepSeek-R1 (Guo et al., 2025), respectively. LIMO is a high-quality distillation dataset containing only 817 examples, where each example contains a question together with the reasoning trajectory and final answer. We provide details of the distillation datasets in Table 4. As described in Appendix B.3, we present the illustration of a sample comprising a question, corresponding reasoning trajectories, and the final answer. In our experiments, following previous work (Muennighoff et al., 2025b; Ye et al., 2025; Liu et al., 2025), we conduct experiments on different base models, including the Qwen2.5-7B-Instruct, Qwen2.5-14B-Instruct and Qwen2.5-32B-Instruct models (Qwen et al., 2025). Specifically, we fine-tune these models on the distillation dataset, enabling models to imitate reasoning trajectories generated by teacher models.

Table 4: Detailed statistics of distillation datasets.

| Datasets | Samples | Avg. Tokens per sample | Domain | Source model | Link |
|---|---|---|---|---|---|
| LIMO (Ye et al., 2025) | 817 | 7126 | Math | QwQ-32B, etc. | Dataset |
| S1 (Muennighoff et al., 2025b) | 1000 | 5058 | Math, Science | Gemini | Dataset |
| S1.1 (Muennighoff et al., 2025a) | 1000 | 10038 | Math, Science | DeepSeek-R1 | Dataset |

### B.2  TRAINING DETAILS

In our experiments, we fine-tune different base models on distillation datasets to obtain distilled reasoning models. In particular, we perform full-parameter fine-tuning on 8 A100 GPUs using DeepSeed ZeRO-3 optimization, with a sequence length limit of 16,384 tokens. In Table 5, we report detailed training parameters employed in the supervised fine-tuning process.

Table 5: Hyperparameters used for supervised fine-tuning.

| Hyperparameter | S1 | S1.1 | LIMO |
|---|---|---|---|
| sequence length | 16384 | 16384 | 16384 |
| batch size | 8 | 8 | 8 |
| learning rate | 1e-5 | 1e-5 | 1e-5 |
| epochs | 5 | 5 | 5 |
| scheduler | cosine | cosine | cosine |
| weight decay | 1e-4 | 1e-4 | 1e-4 |
| zero optimization | ZeRO-3 | ZeRO-3 | ZeRO-3 |
| training precision | bf16 | bf16 | bf16 |

B.3   IMPLEMENTATION EXAMPLES

We provide examples of distillation data and question prompts in Example B.1 and Example B.2. A datapoint from the S1 distillation dataset (Muennighoff et al., 2025b) is presented in Example B.1, comprising a question, corresponding reasoning trajectories, and final answer. In Example B.2, we present three question prompts using the system template.

---

**Example B.1: Illustrative case of distillation data**

**Question:**

Given a rational number, write it as a fraction in lowest terms and calculate the product of the resulting numerator and denominator. For how many rational numbers between 0 and 1 will 20! be the resulting product?

---

**Thinking Trajectories:**

The problem asks for the number of rational numbers between 0 and 1 such that when the rational number is written as a fraction in lowest terms, the product of the numerator and the denominator is 20!.

Let the rational number be $r$, where $0 < r < 1$. We can write $r$ as a fraction $\frac{a}{b}$, where $a$ and $b$ are positive integers, and $\gcd(a, b) = 1$. Since $0 < r < 1$, we have $0 < \frac{a}{b} < 1$, which implies $0 < a < b$. The product of the numerator and the denominator of the fraction in lowest terms is $a \times b$. We are given that $a \times b = 20!$. We need to find the number of pairs $(a, b)$ such that:
1. $a$ and $b$ are positive integers.
2. $\gcd(a, b) = 1$.
3. $a < b$.
4. $a \times b = 20!$.
Let $N = 20!$. We have $a \times b = N$. Since $\gcd(a, b) = 1$, the prime factors of $N$ are distributed between $a$ and $b$ such that no prime factor is shared by both. Consider the prime factorization of $N = 20!$. Let $N = p_1^{e_1} p_2^{e_2} \cdots p_k^{e_k}$, where $p_i$ are distinct prime numbers and $e_i \geq 1$.

Since $a \times b = N$ and $\gcd(a, b) = 1$, each prime power $p_i^{e_i}$ must be entirely a factor of $a$ or entirely a factor of $b$. For each prime power $p_i^{e_i}$, there are two choices: either $p_i^{e_i}$ is a factor of $a$, or $p_i^{e_i}$ is a factor of $b$. Since there are $k$ distinct prime factors, there are $2^k$ ways to distribute the prime power factors between $a$ and $b$. Each way corresponds to a unique pair $(a, b)$ such that $a \times b = N$ and $\gcd(a, b) = 1$. Let the number of distinct prime factors of $N = 20!$ be $\omega(20!)$. Then the number of pairs $(a, b)$ such that $a \times b = 20!$ and $\gcd(a, b) = 1$ is $2^{\omega(20!)}$. Now, we need to consider the condition $a < b$. Since $a \times b = N$, if $a \neq b$, then either $a < b$ or $b < a$. If $a = b$, then $a^2 = N$, which means $N$ is a perfect square. 20! is not a perfect square because the exponent of some prime factors in its prime factorization is odd (e.g., the exponent of the largest prime less than or equal to 20, which is 19). The pairs $(a, b)$ such that $a \times b = N$ and $\gcd(a, b) = 1$ can be grouped into pairs of the form $(a, b)$ and $(b, a)$, where $a \neq b$. The number of such pairs is $2^{\omega(N)}$. Each pair contributes to either $a < b$ or $b < a$. Since $a \neq b$, exactly half of these pairs will satisfy $a < b$.

The number of pairs $(a, b)$ such that $a \times b = N$, $\gcd(a, b) = 1$, and $a < b$ is $\frac{1}{2} \times 2^{\omega(N)} = 2^{\omega(N)-1}$. We need to find the number of distinct prime factors of 20!. The prime numbers less than or equal to 20 are 2, 3, 5, 7, 11, 13, 17, 19. There are 8 distinct prime factors of 20!. So, $\omega(20!) = 8$. The number of pairs $(a, b)$ satisfying the conditions is $2^{8-1} = 2^7 = 128$. Each such pair $(a, b)$ corresponds to a unique rational number $\frac{a}{b}$ between 0 and 1 in lowest terms, where $a \times b = 20!$. The number of rational numbers between 0 and 1 for which the product of the numerator and denominator of the fraction in lowest terms is 20! is 128. Let's verify the reasoning. Consider a simpler example: Product is 12. Prime factorization of 12 is $2^2 \times 3^1$. Distinct prime factors are 2 and 3. $\omega(12) = 2$. Pairs $(a, b)$ such that $a \times b = 12$ and $\gcd(a, b) = 1$: Factors of 12 are 1, 2, 3, 4, 6, 12. Possible pairs $(a, b)$ with $a \times b = 12$: (1, 12), (2, 6), (3, 4). Check $\gcd(a, b) = 1$: $\gcd(1, 12) = 1$ $\gcd(2, 6) = 2 \neq 1$ $\gcd(3, 4) = 1$
The pairs $(a, b)$ such that $a \times b = 12$ and $\gcd(a, b) = 1$ are formed by distributing the prime factors $2^2$ and $3^1$. $a$ can have factors from $\{1, 2^2, 3^1, 2^2 \times 3^1\}$ and $b$ will have the remaining

---

factors. Case 1: $a = 1, b = 12$. $\gcd(1, 12) = 1$. Case 2: $a = 4, b = 3$. $\gcd(4, 3) = 1$. Case 3: $a = 3, b = 4$. $\gcd(3, 4) = 1$. Case 4: $a = 12, b = 1$. $\gcd(12, 1) = 1$.

Something is wrong here. Let's use the prime factorization directly. $12 = 2^2 \times 3^1$. Prime factors are 2 and 3. Possible distributions of prime powers:- $a$ has $2^0, 3^0$, $b$ has $2^2, 3^1 \implies a = 1, b = 12$. $\gcd(1, 12) = 1$.- $a$ has $2^2, 3^0$, $b$ has $2^0, 3^1 \implies a = 4, b = 3$. $\gcd(4, 3) = 1$.- $a$ has $2^0, 3^1$, $b$ has $2^2, 3^0 \implies a = 3, b = 4$. $\gcd(3, 4) = 1$.- $a$ has $2^2, 3^1$, $b$ has $2^0, 3^0 \implies a = 12, b = 1$. $\gcd(12, 1) = 1$. The pairs $(a, b)$ such that $a \times b = 12$ and $\gcd(a, b) = 1$ are (1, 12) and (4, 3) and (3, 4) and (12, 1). Number of such pairs is $2^{\omega(12)} = 2^2 = 4$. Now, consider the condition $a < b$. Pairs where $a < b$: (1, 12) and (3, 4). Number of such pairs is $\frac{1}{2} \times 4 = 2$. The rational numbers are $\frac{1}{12}$ and $\frac{3}{4}$. Product of numerator and denominator: $1 \times 12 = 12$, $3 \times 4 = 12$. Fractions are in lowest terms. Rational numbers are between 0 and 1. The number of distinct prime factors of 20! needs to be correctly calculated. Prime numbers $\leq 20$: 2, 3, 5, 7, 11, 13, 17, 19. There are 8 distinct prime factors. Let $N = 20!$. We are looking for rational numbers $\frac{a}{b}$ such that $\gcd(a, b) = 1$, $0 < a < b$, and $a \times b = N$. The number of pairs $(a, b)$ such that $a \times b = N$ and $\gcd(a, b) = 1$ is $2^{\omega(N)}$. Each pair corresponds to a factorization of $N$ into two coprime factors. Let $N = p_1^{e_1} p_2^{e_2} \cdots p_k^{e_k}$. $a = p_1^{\alpha_1} p_2^{\alpha_2} \cdots p_k^{\alpha_k}$ $b = p_1^{\beta_1} p_2^{\beta_2} \cdots p_k^{\beta_k}$

Since $\gcd(a, b) = 1$, for each $i$, either $\alpha_i = e_i, \beta_i = 0$ or $\alpha_i = 0, \beta_i = e_i$. This gives $2^k$ choices for the pair $(a, b)$. We need $a < b$. Consider the case $a = b$. This happens only if all prime exponents $e_i$ are even, which is not the case for 20!. So, for every pair $(a, b)$ with $a \times b = N$ and $\gcd(a, b) = 1$, we have $a \neq b$. The pairs can be grouped into $(a, b)$ and $(b, a)$, where $a \neq b$. The number of such pairs is $2^{\omega(N)}$. The number of pairs with $a < b$ is $\frac{1}{2} \times 2^{\omega(N)} = 2^{\omega(N)-1}$. For $N = 20!$, $\omega(N) = 8$. The number of pairs $(a, b)$ is $2^{8-1} = 2^7 = 128$. Each pair $(a, b)$ corresponds to a unique rational number $\frac{a}{b}$ between 0 and 1 in lowest terms, such that $a \times b = 20!$. Double-check the problem statement and constraints. Rational numbers between 0 and 1. Resulting product of numerator and denominator is 20!. Consider a rational number $r$ between 0 and 1. When written as a fraction in lowest terms, let it be $\frac{a}{b}$, where $\gcd(a, b) = 1$ and $0 < a < b$. We are given $a \times b = 20!$. We need to find the number of such fractions $\frac{a}{b}$. This is equivalent to finding the number of pairs $(a, b)$ satisfying the conditions. The number of distinct prime factors of 20! is the number of primes less than or equal to 20, which is 8. The primes are 2, 3, 5, 7, 11, 13, 17, 19. Let the prime factorization of 20! be $p_1^{e_1} p_2^{e_2} \cdots p_8^{e_8}$. $a \times b = p_1^{e_1} p_2^{e_2} \cdots p_8^{e_8}$. Since $\gcd(a, b) = 1$, for each prime factor $p_i$, either $p_i^{e_i}$ is a factor of $a$ or $p_i^{e_i}$ is a factor of $b$.

Consider the divisors of 20! that are coprime to their cofactors in the division. If $d$ is a divisor of $N$, then $N = d \times \frac{N}{d}$. We need $\gcd(d, \frac{N}{d}) = 1$. This means that the prime factors of $d$ and $\frac{N}{d}$ are disjoint. Equivalently, for each prime factor $p_i$ of $N$, either $p_i^{e_i}$ divides $d$ or $p_i^{e_i}$ divides $\frac{N}{d}$. Let $a$ be a divisor of 20! such that $\gcd(a, \frac{20!}{a}) = 1$. Then $b = \frac{20!}{a}$. The number of such divisors $a$ is $2^{\omega(20!)}$. These divisors correspond to the possible values of the numerator $a$ in the pairs $(a, b)$ where $a \times b = 20!$ and $\gcd(a, b) = 1$. The pairs are formed by choosing, for each prime power $p_i^{e_i}$, whether it belongs to $a$ or $b$. Example: $12 = 2^2 \times 3^1$. Prime powers are $2^2$ and $3^1$. Choices for factors of $a$: $\{1, 4\}$, choices for factors of $b$: $\{1, 3\}$. Possible values of $a$: $1 \times 1 = 1, 1 \times 3 = 3, 4 \times 1 = 4, 4 \times 3 = 12$. If $a = 1$, $b = 12$. If $a = 3$, $b = 4$. If $a = 4$, $b = 3$. If $a = 12$, $b = 1$. The pairs $(a, b)$ such that $a \times b = N$ and $\gcd(a, b) = 1$ are formed by partitioning the set of prime power factors into two sets, one for $a$ and one for $b$. Let the prime power factors be $P_1, P_2, \ldots, P_k$, where $P_i = p_i^{e_i}$. $a$ is a product of some of these $P_i$, and $b$ is the product of the remaining ones. For each $P_i$, it either goes to $a$ or $b$. $2^k$ possibilities.

We need $a < b$. Since $a \times b = N$, and $a \neq b$ (as $N$ is not a perfect square), either $a < b$ or $b < a$. The number of pairs with $a < b$ is equal to the number of pairs with $b < a$. Total number of pairs is $2^k$. Number of pairs with $a < b$ is $\frac{2^k}{2} = 2^{k-1}$. Here $k = \omega(20!) = 8$. Number of pairs $(a, b)$ with $a < b$, $\gcd(a, b) = 1$, $a \times b = 20!$ is $2^{8-1} = 2^7 = 128$. Each such pair $(a, b)$ corresponds to a unique rational number $\frac{a}{b}$ between 0 and 1 in lowest terms, with the product of numerator

and denominator equal to 20!. Example revisited: Product is 12. Prime factors 2, 3. $\omega(12) = 2$. Pairs $(a, b)$ with $a \times b = 12, \gcd(a, b) = 1$: $a = 1, b = 12$ $a = 3, b = 4$ $a = 4, b = 3$ $a = 12, b = 1$ Pairs with $a < b$: $(1, 12), (3, 4)$. Number $= 2^{2-1} = 2^1 = 2$. Rational numbers: $\frac{1}{12}, \frac{3}{4}$. Consider the edge cases or potential pitfalls. The condition "between 0 and 1" ensures $a < b$. The condition "fraction in lowest terms" ensures $\gcd(a, b) = 1$. The problem boils down to finding the number of ways to factor 20! into two coprime factors $a$ and $b$ such that $a < b$. Final confirmation of the number of distinct prime factors of 20!: Primes $\leq 20$: 2, 3, 5, 7, 11, 13, 17, 19. There are 8 distinct prime factors. The number of rational numbers is the number of pairs $(a, b)$ satisfying the conditions, which is $2^{\omega(20!)-1} = 2^{8-1} = 2^7 = 128$. The question asks for "how many rational numbers", so the answer should be the number of such fractions. Each valid pair $(a, b)$ corresponds to one rational number. If the product was a perfect square, say $36$. $36 = 2^2 \times 3^2$. $\omega(36) = 2$. Pairs $(a, b)$ with $a \times b = 36, \gcd(a, b) = 1$: $a=1, b=36$ $a = 4, b = 9$ $a = 9, b = 4$ $a = 36, b = 1$ Here $a \neq b$. If the product was $p^e$, where $p$ is prime. Pairs $(1, p^e)$ and $(p^e, 1)$. If $e > 0, 1 < p^e$. Rational number $\frac{1}{p^e}$. If the product was $pq$, where $p, q$ are distinct primes. Pairs $(1, pq), (p, q), (q, p), (pq, 1)$. If $p < q$, then $p < q, 1 < pq$. Pairs with $a < b$: $(1, pq), (p, q)$. Number $= 2^{2-1} = 2$. Rational numbers: $\frac{1}{pq}, \frac{p}{q}$. The reasoning seems consistent. The number of rational numbers is indeed $2^{\omega(20!)-1}$.

---

**Answer:**

Solution: Let the rational number between 0 and 1 be $r$. When written as a fraction in lowest terms, let it be $\frac{a}{b}$, where $a$ and $b$ are positive integers, $\gcd(a, b) = 1$, and $0 < a < b$. We are given that the product of the numerator and denominator is 20!, so $a \times b = 20!$. We need to find the number of pairs $(a, b)$ such that $a \times b = 20!, \gcd(a, b) = 1$, and $a < b$. Let $N = 20!$. The prime factorization of $N$ is $p_1^{e_1} p_2^{e_2} \cdots p_k^{e_k}$, where $p_i$ are distinct prime numbers. The number of distinct prime factors of 20! is the number of primes less than or equal to 20. These primes are 2, 3, 5, 7, 11, 13, 17, 19. Thus, there are 8 distinct prime factors, so $k = 8$.

Since $a \times b = N$ and $\gcd(a, b) = 1$, each prime power factor $p_i^{e_i}$ of $N$ must be entirely a factor of $a$ or entirely a factor of $b$. The number of ways to distribute the prime power factors between $a$ and $b$ is $2^k$, where $k$ is the number of distinct prime factors of $N$. Each distribution corresponds to a unique pair $(a, b)$ such that $a \times b = N$ and $\gcd(a, b) = 1$. The number of such pairs is $2^{\omega(N)}$, where $\omega(N)$ is the number of distinct prime factors of $N$.

Since 20! is not a perfect square, for every pair $(a, b)$ such that $a \times b = 20!$ and $\gcd(a, b) = 1$, we have $a \neq b$. The pairs can be grouped into $(a, b)$ and $(b, a)$, where $a \neq b$. The number of pairs with $a < b$ is equal to the number of pairs with $b < a$. The total number of pairs is $2^{\omega(20!)}$. The number of pairs with $a < b$ is $\frac{1}{2} \times 2^{\omega(20!)} = 2^{\omega(20!)-1}$.

The number of distinct prime factors of 20! is $\omega(20!) = 8$. The number of pairs $(a, b)$ such that $a \times b = 20!, \gcd(a, b) = 1$, and $a < b$ is $2^{8-1} = 2^7 = 128$. Each such pair $(a, b)$ corresponds to a unique rational number $\frac{a}{b}$ between 0 and 1 in lowest terms, with the product of the numerator and denominator equal to 20!.

Final Answer: The final answer is $\boxed{128}$

---

## Example B.2: Illustrative case of prompts

**Prompt1:**

```
<|im_start|>system
You are Qwen, created by Alibaba Cloud. You are a helpful assistant. <|im_end|>
<|im_start|>user
Given a rational number, write it as a fraction in lowest terms and calculate
the product of the resulting numerator and denominator. For how many rational
numbers between 0 and 1 will $20_{}^{}!$ be the resulting product?<|im_end|>
<|im_start|>assistant
```

**Prompt2:**

```
<|im_start|>system
You are Qwen, created by Alibaba Cloud. You are a helpful assistant.<|im_end|>
<|im_start|>user
Joanie takes a $\$6,\!000$ loan to pay for her car. The annual interest rate on
the loan is $12\%$. She makes no payments for 4 years, but has to pay back all
the money she owes at the end of 4 years. How much more money will she owe if
the interest compounds quarterly than if the interest compounds annually?
Express your answer as a dollar value to the nearest cent.<|im_end|>
<|im_start|>assistant
```

**Prompt3:**

```
You are Qwen, created by Alibaba Cloud. You are a helpful assistant.<|im_end|>
<|im_start|>user
Gary purchased a large beverage, but only drank $m/n$ of it, where $m$ and $n$
are relatively prime positive integers. If he had purchased half as much and
drunk twice as much, he would have wasted only $2/9$ as much beverage. Find $m+n$
.<|im_end|>
<|im_start|>assistant
```

## C  ADDITIONAL RESULTS

### C.1  TOKEN-WISE PROBABILITY DISTRIBUTION

To compare the probability distribution differences in generated tokens, we analyze the first 300 generated tokens and the generated tokens at positions 700–1000 for both members and non-members. Figure 6 shows the token-wise probability distributions from the distilled reasoning model across 20 member and 20 non-member samples. We find the generated tokens at positions 700–1000 exhibit only minor probability distribution differences between members and non-members (Right). This pattern is significantly different from the generation behavior in the first 300 generated tokens (Left), where members tend to produce high-probability tokens, whereas non-members induce more low-probability tokens. The disparity indicates that earlier generated tokens are likely to exhibit distinct membership signals for members and non-members.

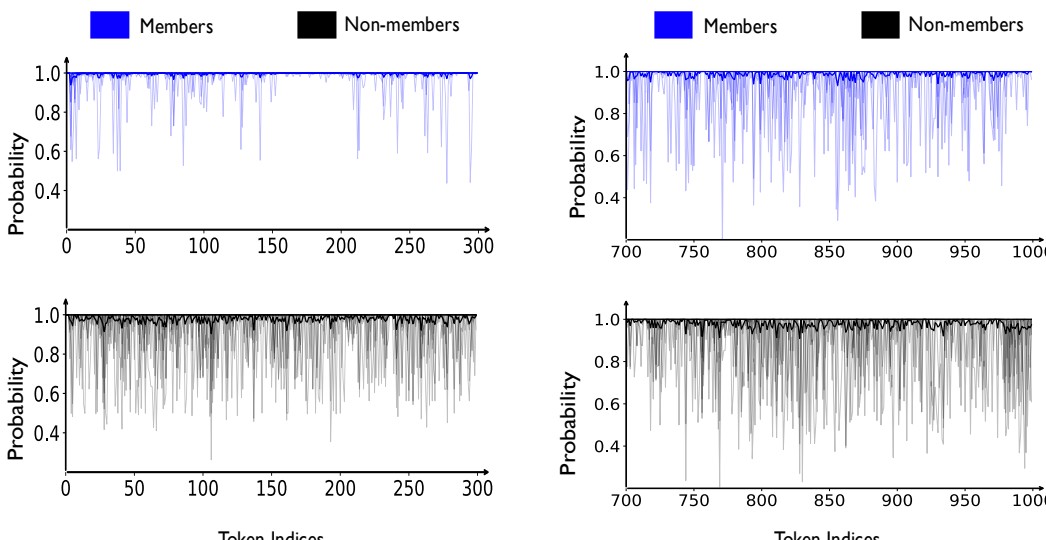

Figure 6: Comparison of token-level generation behaviour of distilled models for 20 member and 20 non-member questions under greedy decoding. **(Left) Token-wise probability distributions of the first 300 generated tokens**: we contrast the distribution of token-wise probability between members and non-members over the first 300 generated tokens. **(Right) Token-wise probability distributions of generated tokens at indices 700 to 1000**: we compare the distribution of token-wise probability between members and non-members over generated tokens at indices 700 to 1000.

## C.2 EXPERIMENTAL RESULTS

In this subsection, we report the additional experimental results. Here, these results are consistent with the conclusions drawn in the main text. Table 6 reports the TPR@1%FPR of our method and baselines on diverse distilled reasoning models. The results show that our method is effective on various dataset across three different models. Table 7 show the AUC and TPR@1%FPR scores of our method and baseline across various models. The experimental results demonstrate that our method is model-agnostic. Table 8 shows the AUC scores of our method and baselines on the S1 dataset across three distinct settings. The results indicate the effectiveness of our method in settings where only the question, or the question along with the reasoning trajectories, is available.

Table 6: TPR@1%FPR of our method and baselines on diverse distilled models. These models are produced through fine-tuning diverse different-sized models (e.g., Qwen2.5-32B-Instruct) on various distillation datasets, including S1, LIMO and S1.1 datasets. † indicates methods that compute score using output tokens. **Bold** shows the superior result.

| Method | Qwen2.5-7B-Instruct | | | Qwen2.5-14B-Instruct | | | Qwen2.5-32B-Instruct | | |
|---|---|---|---|---|---|---|---|---|---|
| | S1 | LIMO | S1.1 | S1 | LIMO | S1.1 | S1 | LIMO | S1.1 |
| *Input-token-based methods* | | | | | | | | | |
| Perplexity (Li, 2023) | 0.040 | 0.000 | 0.005 | 0.015 | 0.012 | 0.005 | 0.015 | 0.006 | 0.000 |
| Lowercase (Carlini et al., 2021) | 0.020 | 0.006 | 0.010 | 0.015 | 0.037 | 0.000 | 0.000 | 0.000 | 0.000 |
| Zlib (Carlini et al., 2021) | 0.025 | 0.000 | 0.000 | 0.015 | 0.006 | 0.000 | 0.005 | 0.000 | 0.000 |
| Neighbor (Mattern et al., 2023) | 0.025 | 0.018 | 0.005 | 0.030 | 0.006 | 0.005 | 0.025 | 0.012 | 0.005 |
| MIN-K% (Shi et al., 2024) | 0.040 | 0.000 | 0.005 | 0.010 | 0.012 | 0.010 | 0.015 | 0.006 | 0.000 |
| MIN-K%++ (Zhang et al., 2025b) | 0.040 | 0.018 | 0.025 | 0.000 | 0.024 | 0.010 | 0.025 | 0.006 | 0.000 |
| Infilling Score (Raoof et al., 2025) | 0.010 | 0.006 | 0.015 | 0.020 | 0.000 | 0.045 | 0.025 | 0.018 | 0.000 |
| *Output-token-based methods* | | | | | | | | | |
| Generated Perplexity† | 0.235 | 0.128 | 0.070 | 0.350 | 0.171 | 0.045 | 0.160 | 0.226 | 0.080 |
| Generated MIN-K%† | 0.235 | 0.128 | 0.070 | 0.350 | 0.171 | 0.045 | 0.160 | 0.226 | 0.080 |
| Ours† | **0.345** | **0.256** | **0.095** | **0.375** | **0.226** | **0.090** | **0.470** | **0.335** | **0.110** |

Table 7: AUC and TPR@1%FPR scores of our method and baselines on S1 dataset across various models, including Llama-3.1-8B- Instruct, Gemma-7B-it and Mistral-7B-Instruct-v0 models. **Bold** shows the superior result.

| Method | AUC | | | TPR@1%FPR | | |
|---|---|---|---|---|---|---|
| | Llama-3.1-8B | Gemma-7b | Mistral-7B | Llama-3.1-8B | Gemma-7b | Mistral-7B |
| Perplexity (Li, 2023) | 0.529 | 0.537 | 0.549 | 0.015 | 0.015 | 0.005 |
| Lowercase (Carlini et al., 2021) | 0.524 | 0.537 | 0.486 | 0.005 | 0.005 | 0.025 |
| Zlib (Carlini et al., 2021) | 0.539 | 0.533 | 0.547 | 0.020 | 0.025 | 0.010 |
| MIN-K% (Shi et al., 2024) | 0.554 | 0.535 | 0.564 | 0.015 | 0.015 | 0.000 |
| MIN-K%++ (Zhang et al., 2025b) | 0.562 | 0.532 | 0.543 | 0.005 | 0.010 | 0.005 |
| Ours | **0.927** | **0.943** | **0.953** | **0.365** | **0.400** | **0.220** |

## D USE OF LARGE LANGUAGE MODELS

This paper utilizes large language models solely for the purpose of enhancing the clarity and precision of specific sentences, without further use of LLMs for additional purposes.

Table 8: AUC scores of our method and baselines on S1 dataset across three distinct settings: using only the question (Question-Only), using the question along with the reasoning trajectories (Question–CoT ), and using the full sample (Question–CoT-Answer). **Bold** shows the superior result.

| Method | Question-Only | Question–CoT | Question–CoT-Answer |
|---|---|---|---|
| Perplexity (Li, 2023) | 0.444 | 0.972 | 0.988 |
| Lowercase (Carlini et al., 2021) | 0.435 | **0.998** | **1.000** |
| Zlib (Carlini et al., 2021) | 0.474 | 0.940 | 0.966 |
| MIN-K% (Shi et al., 2024) | 0.443 | 0.972 | 0.988 |
| MIN-K%++ (Zhang et al., 2025b) | 0.472 | 0.704 | 0.723 |
| Ours | **0.855** | 0.872 | 0.872 |

