# OpenReview forum: "Detecting Distillation Data from Reasoning Models"
_ICLR.cc/2026/Conference — Submitted to ICLR 2026_

### Official Review · Reviewer_gzPs · 2025-10-15

**Soundness:** 3
**Presentation:** 3
**Contribution:** 3
**Rating:** 6
**Confidence:** 5

**Summary:**

The paper addresses the problem of detecting distilled reasoning data, where reasoning chains from a LLM are distilled into a small language model (SLM). The authors aim to identify such distilled data given access to a fine-tuned SLM because some of this data may also appear in benchmark datasets, potentially contaminating the evaluation. They propose a method (TBD) and claim that for questions seen during SLM training, the model generates nearly deterministic answers with very high probabilities. By analyzing these probabilities, it is possible to determine whether a question was included in the SLM’s training data.

**Strengths:**

* Novelty compared to previous methods.
* Simple and intuitive approach.
* Clear mathematical formulas.

**Weaknesses:**

* Focuses on a single LLM (Qwen), though the method could potentially be applied to other models as well.

**Questions:**

* Do you have any evidence to support the claim: "Considering that the earlier generated tokens are likely to be more representative of the behaviour of the model for members and non-members"?

* The difference between your method and others in TPR@1%FPR appears unusually large. Do you have any intuition or explanation for this?

* You state: "In addition, among three models of different sizes, our method achieves the best AUC and TPR@1%FPR on the 32B model, likely because larger models are capable of memorizing training data." However, in Table 1, your method achieves better results with Qwen2.5-7B-Instruct than Qwen2.5-14B-Instruct on LIMO and S1.1. Can you clarify this discrepancy?

* You mention: "The figure 5a shows that the AUC initially increases with truncation length, but declines as the length continues to increase. Our method consistently reaches its optimal performance across various datasets when the truncation length is around 300. Similarly, Figure 5b show that the TPR@1%FPR score exhibits a similar trend. These findings indicate the robustness of our method across various datasets, allowing users to deploy our algorithm with a fixed truncation length value." Why does this behaviour occur? Why is the trend not monotonic? Do you have an intuition for this pattern?

* In Figure 5c, why does AUC behave this way with respect to $\tau$? Increasing $\tau$ allows tokens with higher probability to contribute, which should highlight member samples. On the other hand, the score for classifying a member should be low, but increasing $\tau$ seems to worsen the score (as it increases even for small values). Could you please explain this phenomenon?

---

> ### Author Response · Authors · 2025-11-22
>
> Thanks for your review and the valuable suggestions. Please find our response below.
>
> ### **1. Experiments on different models. [W1]**
> Thank you for the valuable comment about our method's performance on different models. We evaluate our method on the S1 dataset using three different models, including Mistral-7B-Instruct-v0.3, Llama-3.1-8B-Instruct and gemma-7b-it. The results demonstrate the effectiveness of our method across different model architectures. Please find the detailed results in the **General Response**.
>
> ### **2. Evidence for the generation behavior of early-generated tokens. [Q1]**
> Thank you for the question. In Appendix C.1, we add a more extensive analysis by comparing the first 300 generated tokens to the generated tokens between 700–1000 for both members and non-members. Figure 6 illustrates the token-wise probability distributions from the distilled reasoning model across 20 member and 20 non-member samples. The results show that the generated tokens at positions 700–1000 exhibit only **minor** probability distribution differences between members and non-members. This pattern is significantly different from the generation behaviour in the first 300 generated tokens, where members tend to produce high-probability tokens, whereas non-members produce more low-probability tokens. The disparity indicates that earlier generated tokens are likely to exhibit distinct membership signals for members and non-members.
>
>
> ### **3. Explanation for the large TPR@1% FPR gap. [Q2]**
> Thank you for raising the concern about the TPR@1%FPR score. Our method can detect distillation with a high AUC score, which can potentially contribute to improving the TPR@1% FPR. In addition, our method introduces a tunable parameter $\alpha$ to adjust the contribution of tokens to the sample score. An appropriate $\alpha$ (e.g., 0.6) can increase the score of non-members that have low scores and lie close to members in score distribution, thereby enlarging the gap between members and non-members. This strategy further improves the TPR@1% FPR score of our method. In contrast, baseline methods mostly fail to detect distillation data, naturally leading to a very low TPR@1%FPR. Consequently, these results lead to a large gap in TPR@1% FPR between the baselines and our method.
>
> ### **4. Clarification on the discrepancy of our method across different-sized models. [Q3]**
> Thank you for pointing out the potential confusion. Indeed, as shown in Table 1, our method achieves slightly better performance on LIMO and S1.1 with Qwen2.5-7B-Instruct than Qwen2.5-14B-Instruct, while obtaining different conclusion on S1. Therefore, we cannot make a conclusion due to the inconsistent performance across different datasets. It might be because the performance difference between the 7B and 14B models is **modest** compared to the much larger gap when scaling to 32B. Therefore, detection performance is likely influenced by other factors—such as dataset characteristics and training configurations—rather than being determined solely by model capacity.
>
> ### **5. The effect of the truncation length on our method. [Q4]**
> Thank you for the question. Figure 5a shows that AUC initially improves and then drops with longer truncation lengths. This occurs because (1) detection requires enough tokens to capture membership signals, and (2) overly long sequences introduce noise from later tokens, which—as noted in our response to Q1—carry weaker and less discriminative signals. This explains why the non-monotonic behaviour happens.
>
> ### **6. The effect of $τ$ on member scores. [Q5]**
>
> We guess this concern is about how increasing $\tau$ affects the scores of member samples. The table below reports the average sample scores of members and non-members across varying $\tau$. The table shows that increasing  $\tau$ leads to lower scores for members and non-members. This is because increasing $\tau$ includes more high-probability tokens in the score computation, and averaging over these small deviations results in lower sample scores. Notably, the **relative gap** between member and non-member scores increases steadily as $\tau$ increases, producing a more distinct separation between members and non-members. As a result, the AUC of our method improves as $\tau$ increases.
>
>
> | $\tau$  | mean member scores | mean nonmember scores |
> | -------- | -------- | -------- |
> | 0.95  | 0.201    | 0.335     |
> | 0.96  | 0.195    | 0.329     |
> | 0.97  | 0.189    | 0.323     |
> | 0.98  | 0.174    | 0.308     |
> | 0.99  | 0.153    | 0.283     |
> | 1.00  | 0.024    | 0.074     |

---

### Official Review · Reviewer_DBzZ · 2025-10-21

**Soundness:** 3
**Presentation:** 3
**Contribution:** 3
**Rating:** 8
**Confidence:** 4

**Summary:**

This paper addresses the problem of distillation data detection and the key challenge is partial availability: an auditor can query a distilled model with a question, but does not have access to the data used during training. The authors observe that distilled models tend to generate tokens with near-deterministic, high probability for seen questions during distillation, while producing more low-probability tokens for unseen questions. Based on this, they propose Token Probability Deviation (TBD), a scoring function that quantifies how far the probabilities of the first M generated tokens deviate from a high reference probability. A lower TBD score indicates a higher likelihood that the question was in the distillation set. Extensive experiments on models distilled from S1, S1.1, and LIMO datasets show that TBD significantly outperforms existing input-token-based membership inference methods, achieving an AUC of 0.918 and a TPR@1%FPR of 0.470 on the S1 dataset.

**Strengths:**

* The core idea of using the deviation of generated token probabilities instead of input-likelihood is novel
* The experimental evaluation is extensive and TBD consistently outperforms other existing methods across benchmarks
* The good TPR@1%FPR scores also demonstrate its practical utility for real-world auditing
* The paper is clearly written and easy to follow

**Weaknesses:**

* The experiments are conducted based on full access to the model's token-level probability distributions. It would be helpful if the authors could discuss on situations that only the generated text is available.
* It would be interesting to know whether sampling-based decoding strategies, e.g., top-k, temperature score, might affect the TBD scores.

**Questions:**

1. How would the performance of TBD if only generated text is available instead of full token probabilities?
2. Have you explored the performance change of TBD using sampling with different temperature scores?
3. Would it affect the performance of TBD if we change the distillation process from fine-tuning into DPO or RLHF?

---

> ### Author Response · Authors · 2025-11-22
>
> Thanks for your recognition and the valuable suggestions. Please find our response below.
>
> ### **1. Assume access to the model’s token-level probabilities. [W1, Q1]**
> Yes, our method requires access to model outputs' probabilities, which is also necessary in current detection methods for LLMs [1, 2, 3, 4]. Note that current distilled LLMs are generally open-sourced (e.g., DeepSeek-R1-Distill-Qwen-7B, Qwen3-8B, s1-32B and LIMO-32B), making it practical to access the logit outputs. Besides, many API-based LLMs also provide access to token-level probabilities, such as GPT-4o and Deepseek-V3. In summary, the assumption of access to token-level probabilities is realistic and commonly adopted in training data detection. As our method requires access to the probabilities of generated tokens, it is not yet applicable to the setting where only the generated text is available. Importantly, we agree that it will be a promising direction to detect distillation data with only generated texts.
>
> ### **2. Experiments with different sampling strategies. [W2, Q2]**
> Thank you for the valuable question. As suggested, we conduct experiments on the S1 dataset with different sampling parameters under Qwen2.5-7B-Instruct, including temperature (e.g. 0.3, 0.5, 0.7), and top-k (e.g. 10, 30, 50).  For all experiments, we set the top‑p sampling parameter to 0.95. For the experiment with varying temperature, we set the top-k to 30.  For the experiment with varying top-k,  the temperature is fixed at 0.3. The table below presents the AUC score of our method with different sampling parameters. The results show that our method can achieve better performance with a lower temperature, while its performance remains stable across different Top‑k values. Additionally, our method achieves superior performance via greedy decoding.
>
> | Type | Temperature |       |       | Top-k |       |       | Greedy decode |
> | ---- | ----------- | ----- | ----- | ----- | ----- | ----- |:-------------:|
> |      | 0.3         | 0.5   | 0.7   | 10    | 30    | 50    |       \       |
> | Ours | 0.712       | 0.679 | 0.689 | 0.712 | 0.712 | 0.712 |     **0.855** |
>
> ### **3. Results under distillation with RLHF or DPO. [Q3]**
> Thank you for the suggestion. In reasoning distillation, almost **ALL** current works employed full-parameter supervised fine-tuning (SFT) to transfer reasoning abilities to small language models [5, 6, 7, 8]. Thus, our method focuses on the detection task in reasoning distillation with full-parameter SFT. Reviewers might be interested in the detection performance under other training strategies, despite their infrequent use in the current literature. To this end, we conduct new experiments with DPO-based reasoning distillation by training Qwen2.5-7B-Instruct on Math-Reasoning-DPO [9]. The results below show that all methods fail to provide meaningful detection performance in this setting, indicating a new challenge in the community. SFT increases confidence on responses to seen examples while decreasing it for unseen examples, whereas DPO decreases the confidence of almost all responses for both seen and unseen examples [10]. We conjecture that the DPO training paradigm differs fundamentally from SFT, rendering all existing detection signals completely invalid in the new setting. We add this into the limitation in the revised version and believe it will be an interesting direction for future work.
>
> | Method | Perplexity | Lowercase |Zlib | MIN-K%|MIN-K%++|Ours|
> | -------- | -------- | -------- |-------- |-------- | -------- |-------- |
> | AUC  | 0.517 |0.541 | 0.484|0.520|0.520|0.507|
>
> [1] Shi W, et al. Detecting Pretraining Data from Large Language Models, ICLR, 2024.
> [2] Zhang J, et al. Min-k%++: Improved baseline for detecting pre-training data from large language models, ICLR, 2025.
> [3] Raoof N, et al. Infilling Score: A Pretraining Data Detection Algorithm for Large Language   Models ICLR, 2025.
> [4] Mattern J, et al. Membership Inference Attacks against Language Models via Neighbourhood Comparison, ACL 2023.
> [5] Yang A, et al. Qwen3 technical report. arXiv:2505.09388, 2025.
> [6] Guo D, et al. Deepseek-r1: Incentivizing reasoning capability in llms via reinforcement learning. arXiv:2501.12948, 2025.
> [7] Muennighoff N, et al. s1: Simple test-time scaling. EMNLP 2025.
> [8] Ye Y, et al. Limo: Less is more for reasoning. arXiv:2502.03387, 2025.
> [9] Math-Reasoning-DPO: https://huggingface.co/datasets/est-ai/math-reasoning-dpo.
> [10] Ren Y, et al. Learning Dynamics of LLM Finetuning, ICLR, 2025.

---

### Official Review · Reviewer_pJZQ · 2025-11-01

**Soundness:** 2
**Presentation:** 3
**Contribution:** 2
**Rating:** 4
**Confidence:** 3

**Summary:**

The authors propose token probability deviation (TBD), a method for detecting instances which were used to distill reasoning data from long reasoning models (LRMs) into smaller LMs. Te authors propose a metric for detecting data used in training based on low probability tokens generated by the LM post-distillation. Higher likelihood of generating the reasoning sequence indicates that the instance was used in distillation. The authors evaluate the method using three distillation datasets (S1, S1.1 and LIMO) across Qwen models of different sizes, and show that it outperforms input-based methods as well as several baselines on detecting distillation data.

Overall, the paper is very well written and motivated. The authors motivate their approach well using emprical data, as well as describe the methodology clearly. My main issue with the work is that the contribution feels very limited and the method is a straightforward exstension of perplexity - TBD filters out high probability tokens, and scales the (1-) probabilities before averaging them. See questions for other concerns.

**Strengths:**

- Well written and motivated paper
- A novel method that performs well on detecting distillation data

**Weaknesses:**

- The contribution is very limited
- Methodological novelty is very low

**Questions:**

- Did you try paraphrasing the input questions before distillation, and then detecting reasoning? A simple way of bypassing this detection method would seem to generate reasoning for distillation based on paraphrased input, where the model should still perform well on data, but the reasoning chains might differ sufficiently to avoid detection.
- Why are the perplexity and min-k scores computed on the first 1000, while TBD on the first 300 tokens? This makes the comparison unclear, especially since it is shown in Figure 5 that the performance of TBD decreases at higher truncation lengths.

---

> ### Author Response · Authors · 2025-11-22
>
> We thank the reviewer for the comments and questions. Please find our response below.
> ### **1. Clarification on contributions and novelty of our work. [W1, W2]**
> We thank the reviewer for raising concerns about the contributions and novelty of our work. Below, we clarify the key contributions and highlight how our work meaningfully differs from prior approaches.
>
> * **A novel and unexplored task.** We formalize **distillation data detection** — a practical yet underexplored problem, where benchmark questions used in reasoning distillation may contaminate the evaluation data. The unique challenge of this task lies in **partial availability**: only the question is available at detection, without access to reasoning trajectories and answers. Accessing question-response pairs is generally infeasible due to the proprietary nature of datasets and the non-deterministic generation process in solution construction. Existing detection methods rely on full sample information and thus fail in this realistic setting. Our work is the first to explicitly define and tackle detection under this strict constraint.
>
> * **Novel methodology and insight.**  Existing approaches targeting LLMs typically score input sequences, and therefore struggle to identify distillation data under the realistic constraint in which only the question is available. We propose a novel and effective method tailored to distillation data detection, which leverages generated tokens rather than the input sequence to achieve effective detection. Our method is established on the new insight that distilled reasoning models tend to generate tokens with high probability for members, while producing more low-probability tokens for non-members. Our work demonstrates that the generated tokens can serve as membership signals for distillation data detection, a mechanism fundamentally different from previous methods.
>
> In summary, our contributions are twofold: (1) identifying and formalising a novel, practically motivated training data detection task under partial-availability constraints, and (2) introducing a novel and effective method that leverages generated token probability patterns as a new type of membership signals. These elements collectively distinguish our work from previous training data detection techniques for LLMs.
>
> ### **2. Experiments on input question paraphrasing. [Q1]**
> Thank you for the great question. To examine the performance of our method under reasoning distillation with paraphrased questions, we conduct experiments on the S1 dataset across Qwen2.5-7B-Instruct, Qwen2.5-14B-Instruct and Qwen2.5-32B-Instruct models. We use GPT-5-mini to paraphrase the original question, obtaining a rephrased version that remains semantically consistent with the original question. We then evaluate our method on paraphrased questions to simulate a setting in which the original questions used for reasoning distillation are unavailable. The table below reports the AUC scores of baselines and our method. We find that our method consistently outperforms baselines. The results show our method is capable of detecting distillation data in the question paraphrasing setting.
>
> | Method | Qwen2.5-7B-Instruct | Qwen2.5-14B-Instruct |Qwen2.5-32B-Instruct|
> | -------- | -------- | -------- |-------- |
> | Perplexity  | 0.463    | 0.468     | 0.469    |
> | Lowercase  | 0.503    | 0.493     | 0.543    |
> | Zlib  | 0.497     | 0.497    | 0.496     |
> | MIN-K%   | 0.469     | 0.475   | 0.473     |
> | MIN-K%++   |0.500     | 0.494     |0.527  |
> | Ours  | **0.615**    | **0.692**     | **0.691** |
>
> ### **3. Clarification on the truncation lengths used in our method. [Q2]**
> Thank you for raising the concern. We clarify that the two variants ("Generated Perplexity" and "Generated Min-K%") are presented as an **ablation study** to show the effectiveness of specific designs in our method based on the generated information. Thus, we remove the truncation operation from the two variants, since the refined truncation operation is a unique design in our method. However, it is computationally costly to use all model outputs due to their excessive length. Thus, we use a fixed truncation length (i.e., 1000) in the two variants to perform the ablation study.

---

### Official Review · Reviewer_yhoQ · 2025-11-03

**Soundness:** 2
**Presentation:** 3
**Contribution:** 2
**Rating:** 4
**Confidence:** 3

**Summary:**

This work introduces the task of distillation data detection, namely in a question-only setting where one is assessing whether a question was trained on during the distillation process. They propose token probability deviation (TBD) which is a score derived from the probabilities of *generated tokens* during greedy decoding. This method is motivated by an observation that questions that were seen during distillation (members) yield more deterministic output behavior than those not seen (non-members). Testing on three distillation datasets (S1, S1.1, LIMO) and Qwen2.5 models, they fine their  with Qwen2.5 models (7B/14B/32B), the method outperforms input-likelihood and simple output-based baselines on membership identification. They also conduct ablations and sensitivity analysis to understand reasonable gating thresholds and token caps to maximize AUC and TPR@1%FPR.

**Strengths:**

- **Problem formulation makes sense:** Data distillation detection seems like a valid and important task with the growing sigfnificance of distillation methods, particularly for with longer-CoT methods. There are cases where the actual distillation data itself could be unavailable (e.g., proprietary long-CoT outputs for a public task).
- **Intuitive and largely model-agnostic:** It operates on generated-token probabilities and is straightforward to implement, assuming model output probabilities are available.

**Weaknesses:**

- **Limited novelty:** TBD is largely a fairly intuitive heuristic (with a couple of parameters to e.g., gate token length, deviation penalty, etc.), however it seems fairly derivative given existing related work on conventional data contamination methods.
- **Could use a wider variety of model and training configurations**: the work is tested only on QWEN-2.5 with conventional supervised fine-tuning and a small range of dataset sizes. To claim the robustness of this metric, it would be ideal to test this on a wider range of setups and given the metric is the primary contribution of the work, this area needs significantly more robust exploration before I would feel comfortable deploying it for this task.
- **Contribution scope**: While the scope is clear, I feel like this work would benefit from comparative analysis that consider multiple formulations of distillation data detection task -- i.e., one with full distillation information (q,c,a) and one with only the defined partial availability (q). I think it's reasonable to assume there may be parallels between these two formulations and I am not sure if it makes to address just the partial availability setting without performing some empirical comparison with the more conventional 'full availability' setting.

**Questions:**

- Can you provide results for the other popular open-weight model families?
- Similarly, have you explored any more nuanced distillation strategies beyond full-parameter supervised fine-tuning?

It seems these two above dimensions could have a significant impact on distillation detection performance.
- Do you have any baseline performance numbers if conditioning on the full training sample (or perhaps just question+rationale / rationale)? I think some additional analysis in this direction would paint a clearer picture for the distillation data detection task as a whole.

Writing / Grammar Suggestions:
- The acronym for the technique (TBD) does not match the name (Token Probability Deviation)? Is this intentional? If so, perhaps highlight the **b** in probability when introducing the term?
- I would appreciate if you could include a table outlining the key statistics of each distillation dataset and the key reasoning characteristics they are attempting to distill.

---

> ### Author Response · Authors · 2025-11-22
>
> Thanks for your review and the valuable suggestions. Please find our response below.
> ### **1. Clarification on our method's novelty. [W1]**
> We thank the reviewers for raising concerns about the novelty of our work. Below, we clarify the key contributions and highlight how our work meaningfully differs from prior approaches.
>
> * **Novel task.** We formalize **distillation-data detection** — a practical yet underexplored problem, where benchmark questions used in reasoning distillation may contaminate evaluation sets. The unique challenge of this task lies in **partial availability**: only the question is available at detection, without access to reasoning trajectories and answers. Existing detection methods rely on full sample information and thus fail in this realistic setting. Our work is the first to explicitly define and tackle detection under this strict constraint.
>
> * **Novel methodology and insight.** Unlike previous methods that score input sequences, we propose using the probability distribution of self-generated tokens as the membership signal. Our core insight — member questions elicit high-probability continuations while non-members induce lower-probability tokens — is, to our knowledge, entirely new in the training data detection for LLMs.
>
> In summary, our contributions are twofold: (1) identifying and formalizing a new, practically motivated detection task under strict partial-availability constraints, and (2) introducing a simple yet highly effective method that leverages generated-token probability patterns as a new class of membership signals. These elements collectively distinguish our work from previous training data detection techniques for LLMs.
>
> ### **2.Experimental results on different models and training configurations. [W2, Q1]**
> Thank you for raising the concern. We present experimental results for various models and training configurations below.
>
> * **Experiments on different models.** Thank you for the suggestion. We provide new experiments on the S1 dataset using three models, including Mistral-7B-Instruct-v0.3, Llama-3.1-8B-Instruct and gemma-7b-it. The results demonstrate that our method is model-agnostic, showing its effectiveness across various model architectures. Please find the detailed results in the General Response.
>
>  * **Experiments on different training configurations.** We note that the reviewer is concerned about the effectiveness of our method across different training setups. We provide new experiments on the S1 dataset across different training configurations. Concretely, following previous work[1], we fine-tune Qwen2.5-7B-Instruct model with different training parameters, including learning rate (e.g. $1 \times 10^{-6}$, $1 \times 10^{-5}$, $1 \times 10^{-4}$) and epoch (e.g. 3, 4, 5). The table below reports the AUC scores of baselines and our method across different training setups. The results show that **our method consistently outperforms all baselines**, demonstrating the effectiveness of our method across different training setups. We find that the performance of our method improves with more training epochs, while a learning rate of $1 ×10−6$ is too small for the model to converge, leading to suboptimal results. In practice, our method is applicable to real-world scenarios, where model training commonly uses suitable configurations, such as a learning rate of $1 \times 10^{-5}$ and 5 epochs—as in our experiments.
>
> | Type   | Epoch |       |       |   Learning rate     |        |  |
> | ------ | ------------- | ----- | ----- |----- |----- |----- |
> | Method | 3 | 4| 5|$1 \times 10^{-6}$ | $1 \times 10^{-5}$  | $1 \times 10^{-4}$|
>  Perplexity  | 0.448    | 0.447    | 0.444    |  0.453    | 0.444    | 0.546    |
> | Lowercase  |0.435   | 0.434     | 0.435    | 0.432    | 0.435    | 0.524   |
> | Zlib  | 0.477     | 0.476    | 0.474     |  0.477    | 0.474     | 0.549  |
> | MIN-K%   |0.448     | 0.445   |0.443     |  0.453   | 0.443     | 0.572    |
> | MIN-K%++   |0.474    | 0.476     |0.472 |  0.473  | 0.472   | 0.551    |
> | TBD    | **0.579** |**0.753** | **0.855**  |**0.495** |**0.855** | **0.956** |

---

> ### Author Response · Authors · 2025-11-22
>
> ### **3. Access to trajectories or answers. [W3, Q3]**
> Thanks for your valuable suggestions. First, we emphasise that access to only questions is more realistic in reasoning distillation, since accessing question-response pairs is generally infeasible due to the proprietary nature of datasets and the non-deterministic generation process in solution construction. To illustrate the scenarios where our method provides utility, we conduct new experiments on the S1 dataset using Qwen2.5-7B-Instruct under three new settings: (1) using only the question, (2) using the question along with the reasoning trajectories, and (3) using the full sample comprising the question, reasoning trajectories and answer. Our method relies solely on the generated tokens and does not use the provided reference answers across the 3 settings (This explains why our results are the same between the two new settings ).
>
> The table below reports the AUC scores of the compared methods and ours. The results show that our method is the only effective one in the question-only settings, while all previous methods fail. This validates the unique contribution of this work in the realistic setting. When it comes to the settings with access to trajectories or full samples, all methods can achieve excellent performance in the detection task. We conjecture that the trajectories and answers contain lots of membership signals that significantly degrade the task difficulty. In summary, the central contribution of this work is to enable effective detection in the most realistic and challenging setting, where the model is given access to the question alone, achieving meaningful performance without relying on trajectories or answers.
>
> | Methods | Question-only $(q)$| Question-trajectories $(q, c)$ |Full sample $(q, c ,a)$|
> | -------- | -------- | -------- | -------- |
> | Perplexity  | 0.444     | 0.972     | 0.988    |
> | Lowercase  | 0.435     | **0.998**     | **1.000**    |
> | Zlib  | 0.474     | 0.940     | 0.966     |
> | MIN-K%   | 0.443     | 0.972    | 0.988     |
> | MIN-K%++   |0.472     | 0.704     |0.723  |
> | Ours  | **0.855**    | 0.872     | 0.872 |
>
> ### **4. Results on other distillation strategies. [Q2]**
>
> Thank you for the suggestion. In reasoning distillation, most current works employed full-parameter supervised fine-tuning (SFT) to transfer reasoning abilities to small language models [2, 3, 4, 5]. Thus, our method focuses on the detection task in reasoning distillation with full-parameter SFT. Reviewer might be interested in the detection performance under other training strategies, despite their infrequent use in the current literature. To this end, we conduct new experiments with DPO-based reasoning distillation by training Qwen2.5-7B-Instruct on Math-Reasoning-DPO [6]. The results below show that all methods fail to provide meaningful detection performance in this setting, indicating a new challenge in the community. SFT increases confidence on responses to seen examples while decreasing it for unseen examples, whereas DPO decreases the confidence of almost all responses for both seen and unseen examples [7]. We conjecture that the DPO training paradigm differs fundamentally from SFT, rendering all existing detection signals completely invalid in the new setting. Thus, we add this into the limitation in the revised version and believe it will be an interesting direction for future work.
>
>
> | Method | Perplexity | Lowercase |Zlib | MIN-K%|MIN-K%++|Ours|
> | -------- | -------- | -------- |-------- |-------- | -------- |-------- |
> | AUC  | 0.517 |0.541 | 0.484|0.520|0.520|0.507|
>
> ### **5. Suggestions**
> Thank you for the suggestions. We have fixed these in the updated version.
>
> * Yes, the “**b**” in the acronym is intended to represent "pro**b**ability". We make this clearer by explaining the “b” when first introducing the term in the revised version.
> * We have included a table in Appendix B.1 that presents the key statistics and detailed characteristics of each distillation dataset.
>
>
> [1] Muennighoff N, et al. s1: Simple test-time scaling, EMNLP 2025.
> [2] Yang A, et al. Qwen3 technical report. arXiv:2505.09388, 2025.
> [3] Guo D, et al. Deepseek-r1: Incentivizing reasoning capability in llms via reinforcement learning. arXiv:2501.12948, 2025.
> [4] Muennighoff N, et al. s1: Simple test-time scaling. EMNLP 2025.
> [5] Ye Y, et al. Limo: Less is more for reasoning. arXiv:2502.03387, 2025.
> [6] Math-Reasoning-DPO: https://huggingface.co/datasets/est-ai/math-reasoning-dpo.
> [7] Ren Y, et al. Learning Dynamics of LLM Finetuning, ICLR, 2025.

---

### Author Response · Authors · 2025-11-22
**General Response**

# **General Response**
We appreciate the reviewers' thoughtful feedback and valuable comments on our work. We are encouraged that reviewers recognize that the task of distillation data detection we proposed is **important** (yhoQ). The reviewers point out that our method is **intuitive** (yhoQ, gzPs) and **novel** (DBzZ, gzPs) by leveraging generated tokens for detection. We are pleased that the reviewer considers our experiments **extensive** (DBzZ), and the empirical results show **practical utility** for real-world auditing (DBzZ). Besides, reviewers recognize that the writing is **well** (pJZQ) and **easy to follow** (DBzZ), with **clear mathematical formulas** (gzPs).

In the following responses, we responded to each reviewer's comments in detail, respectively. We have revised the manuscript according to reviewers' suggestions, and we believe this makes our paper much stronger. The main changes we made are summarized below:

For clarity, we highlight the revised part of the manuscript in **blue** color.

* Added the introduction of notation in **Line 76**. [yhoQ]
* Added experiments on various models in **Line 374-393** and **Appendix C.2**. [yhoQ, gzPs]
* Revised the description of experimental results in **Line 400-402** and **Appendix C.2**. [gzPs]
* Added experiments on new settings in **Line 502-512** and **Appendix D.3**. [yhoQ]
* Added experiments on question paraphrasing in **Line 512-522** and **Appendix D.3**. [pJZQ]
* Added a table with details of distillation datasets in **Appendix B.1**. [yhoQ]
* Added analysis of probability distribution in generated tokens in  **Appendix C.1**. [gzPs]
---


Here, we provide responses to some common concerns of the reviewers.

### **1. Experiments on various models.**
We note that two reviewers are interested in the effectiveness of our method on various models. We include new experiments on the S1 dataset with three additional LLMs: Llama-3.1-8B-Instruct [1], Gemma-7B-it [2], and Mistral-7B-Instruct-v0.3 [3]. These models were fine-tuned using the identical training setup detailed in Appendix B.2. The table below presents the AUC scores for our method and the baselines across all evaluated models. The results reveal that our method **consistently achieves superior performance compared to all baselines** across various models, clearly highlighting its model-agnostic nature and broad applicability.

| Methods | Llama-3.1-8B-Instruct | Gemma-7b-it |Mistral-7B-Instruct-v0.3|
| -------- | -------- | -------- | -------- |
| Perplexity  | 0.529     | 0.537     | 0.549     |
| Lowercase  | 0.524     | 0.537     | 0.486     |
| Zlib  | 0.539     | 0.533     | 0.547     |
| MIN-K%   | 0.554     | 0.535     | 0.560     |
| MIN-K%++   | 0.562     | 0.532     | 0.540     |
| Ours  | **0.927**     | **0.943**     | **0.953**|

[1] Meta AI. Introducing Llama 3.1: Our most capable models to date. Meta Blog.
[2] Team G, et al. Gemma: Open models based on gemini research and technology. arXiv:2403.08295, 2024.
[3] Albert Q. J, et al. Mistral 7B. arXiv:2310.06825, 2023.

---

### Meta-Review · Area_Chair_SSnv · 2026-01-07

**Summary:**

This paper studies the detection of reasoning distillation. Reasoning distillation is an emerging method for enhancing the reasoning capabilities of LLMs. However, this may cause benchmark contamination. The proposed method is motivated by the following observation: distilled models tend to generate confident tokens for seen questions, while producing low-probability tokens for unseen questions. The paper further performs the experiments to demonstrate the effectiveness of the proposed method.

**Reviewer Concerns:**

The paper has mixed reviews. In general, the reviewers note that the paper is easy to follow and the proposed method is very intuitive. The paper also compares the proposed method with baselines to demonstrate its effectiveness. However, there are also concerns. For instance, the proposed method is based on intuitive heuristics: how far the probabilities of generated tokens are from being fully deterministic. This is an intuitive and simple heuristic. The paper may highlight the novelty of the proposed method over prior studies (especially in membership inference literature). The idea has also been explored to detect AI-generated text (see the related work discussion in DetectGPT: Zero-Shot Machine-Generated Text Detection using Probability Curvature). The paper may perform a comprehensive discussion on the connection of the proposed method with prior studies. Second, the effectiveness of the proposed method may diminish under general settings (as shown in the response). Additionally, the paper may report the false positive rate and false negative rate with a threshold. In practice, we may need to select a threshold to determine if a question is in the distill dataset. The threshold may need to generally work well across different settings (since the distillation dataset is unknown).

**Reviewer Scores:**

The reviewers may not change their scores.

---

### Decision · Program_Chairs · 2026-01-26

Reject